# Analytical expressions for time evolution of spin systems affected by two or more interactions

Günter Hempel[1]

[1]Martin-Luther-Universität Halle-Wittenberg, Institut für Physik – NMR, Betty-Heimann-Str. 7, 06120 Halle, Germany

**Correspondence:** Günter Hempel (guenter.hempel@physik.uni-halle.de)

**Abstract.** Analytical expressions for the description of the time evolution of spin systems beyond the product-operator formalism (POF) can be obtained if a low-dimensional subspace of the Liouville space has been found in which the time evolution of the spin system takes place completely. This can be achieved by a procedure that consists of a repeated application of the commutator of the Hamiltonian with the density operator. This iteration continues as long as the result of such a commutator operation contains a term that is linearly independent of all the operators appearing in the previous commutator operations. The coefficients of the resulting system of commutator relations can be immediately inserted into the generic propagation formulae given in this article if the system contains two, three or four equations. In cases where the validity conditions of any of these propagation formulae are not met, the coefficients are used as intermediate steps to obtain both the Liouvillian and propagator matrices of the system. Several application examples are given where an analytical equation can be obtained for the description of the time evolution of small spin systems under the influence of two or more interactions. This procedure for finding the Liouvillian matrix is not limited to time-independent interactions. Some examples illustrate the treatment of time-dependent problems using this method.

## 1  Introduction

Considerable progress has been made in predicting the time evolution of spin systems. Numerical calculations or simulations are possible in large systems of coupled spins (Kuprov et al., 2007), and the evolution during an arbitrarily long sequence of pulses can be simulated with the help of software (Veshtort and Griffin, 2006; Bak et al., 2000; Hogben et al., 2011), including the effect of thermal motion.

However, there may be situations where the analytical representation of the time evolution is advantageous, providing more physical intuition than a numerical procedure which can seem like a black box. The desire or need to take a look inside this black box could be a motivation to deal with analytical contexts. In many cases, the product-operator formalism (POF) (Slichter, 1987; Sorensen et al., 1983; Packer and Wright, 1983; Wang, 1986) has been used for this purpose. This is a rather intuitive scheme, where the states of the spin system are represented by spin operators. Such a tool can be very useful for doing short calculations without access to fundamental quantum mechanics or without a simulation program. For demonstrations in discussions and lectures, e.g. on the effects of pulse sequences like INEPT, HSQC (vandeVen and Hilbers, 1983), an illustrative explanation can be given here.

As an example, consider the POF description of the propagation of transversal magnetization of spin $I = 1/2$ which is scalarly coupled to another spin $S = 1/2$ (coupling constant $J$):

$$\hat{I}_x \quad \xrightarrow{\quad 2\pi J \hat{I}_z \hat{S}_z \cdot t \quad} \quad \hat{I}_x \cos \pi J t \quad + \quad 2\hat{I}_y \hat{S}_z \sin \pi J t \tag{1}$$

This means that the spin system oscillates between two states characterized by the operators $\hat{I}_z$ and $2\hat{I}_z\hat{S}_z$. Thus, the evolution takes place in a two-dimensional subspace of the total operator space. In contrast, to describe a system with two spins 1/2, we need a $4 \times 4$ density matrix. This means that the Liouville-von Neumann equation is a system of 16 scalar differential equations for each of the matrix elements. But obviously it is possible in this example to reduce the 16D problem to the 2D one described by equation (1).

The question arises: Are there other situations where dimensionality reduction is possible? For example: Evolution of a spin system under dipole-dipole interaction and rf irradiation, or cross polarization regarding the finiteness of rf power, i.e.

$$\hat{I}_z \quad \xrightarrow{\quad \text{dipole-dipole interaction + rf irradiation} \quad} \quad ?$$

$$\hat{I}_x \quad \xrightarrow{\quad \text{cross polarization under finite rf power} \quad} \quad ?$$

or others? For numerical calculations, any reduction in dimensionality leads to a reduction in computational time, but for enabling analytical calculations, which is the goal here, it may be essential. The aims of this paper are (i) to introduce a procedure for reducing the dimension of the problem, (ii) to show examples of the application of this procedure to obtain analytical equations, and (iii) to show the resulting generic propagation rules as templates for the cases that the problem could be reduced to a 2D, 3D, or 4D problem. The latter can be seen as an extension of the product-operator formalism to somewhat more complex situations. (i) and (ii) can be useful for simplifying some calculations by dimension reduction without any approximation. Even for numerical calculations this can be useful if it helps to work on low-dimensional systems. As an example, an IS spin system is mentioned here, which is subject to the di-polar interaction, but which is decoupled by rf irradiation, and at the same time fast sample rotation (MAS) modulating the dipolar oscillation takes place. Its time evolution can be described by a system of 3 differential equations using the method presented here. The application of the Shirley-Floquet method will be greatly simplified here. After all, the application of the Liouville-vonNeumann equa-tion in the four-dimensional wavefunction space leads to a system of 16 differential equations even if some of its coefficients can be zero. In section 2, the mathematical background is investigated, which makes it possible to reduce the dimension to the value 2 in cases where the POF is applicable. The results of this consideration are applied to more complex structures in section 3. Finally, in section 4, some template formulae are given together with examples for cases where a reduction to 3D, 4D, 5D and 6D problems is possible.

As usual, in this manuscript operators are denoted by a hat ($\hat{A}$) and superoperators by a double hat ($\hat{\hat{B}}$), while the vector or the matrix associated with this operator is denoted by the same symbol but in bold style and without hat ($\mathbf{A}$, $\mathbf{B}$). Scalar variables are written in italics.

## 2 Dimension reduction through POF

### 2.1 Condition for validity: Commutator relations

As shown in the references cited above, the generic scheme of the POF is as follows: The time evolution can be predicted by the propagation rules

$$\hat{A} \quad \xrightarrow{\hat{H}\,t} \quad \hat{A}\cos\lambda t + \hat{B}\sin\lambda t \quad ;$$
$$\hat{B} \quad \xrightarrow{\hat{H}\,t} \quad \hat{B}\cos\lambda t - \hat{A}\sin\lambda t \tag{2}$$

if and only if

$$\left[\hat{H},\hat{A}\right] = \quad i\lambda\hat{B} \quad \text{and}$$
$$\left[\hat{H},\hat{B}\right] = -i\lambda\hat{A} \tag{3}$$

where $\left[\hat{A},\hat{B}\right] \equiv \hat{A}\hat{B} - \hat{B}\hat{A}$ denotes the commutator between the operators $\hat{A}$ and $\hat{B}$. That is, Eq. (2) describes the motion of the density operator in a two-dimensional subspace of the total Liouville space of the current spin system although the Liouville space has a much larger dimension.

However, condition (3) is often not satisfied when more than one interaction must be considered. This applies, for example, to an rf irradiation with strength $\omega_1$ on an ensemble of spins $I = 1/2$ which are coupled to another spin $S = 1/2$ (coupling frequency $D_{IS}$). This situation can be described by the Hamiltonian $\hat{H} = -2D_{IS}\hat{I}_z\hat{S}_z - \omega_1\hat{I}_x$ and an initial state $\rho_0 = \hat{I}_z$. The double calculation of the commutator of the Hamiltonian with $\hat{I}_z$ does not only result in $\hat{I}_z$, but an additional term appears. In this case, after the third application of this commutator operation, the result will contain only the operators already used:

$$\left[\hat{H},\hat{I}_z\right] = \quad i\omega_1\,\hat{I}_y$$
$$\left[\hat{H},\hat{I}_y\right] = -iD_{IS} \cdot 2\hat{I}_x\hat{S}_z - i\omega_1\,\hat{I}_z$$
$$\left[\hat{H},2\hat{I}_x\hat{S}_z\right] = \quad iD_{IS} \cdot \hat{I}_y \tag{4}$$

The connection between the commutator equations (3) and the propagation formulae (2) becomes much clearer if we reformulate both sets of equations in matrix form:

$$\begin{pmatrix} \hat{A} \\ \hat{B} \end{pmatrix} \quad \xrightarrow{\hat{H}\cdot t} \quad \begin{pmatrix} \cos\lambda t & \sin\lambda t \\ -\sin\lambda t & \cos\lambda t \end{pmatrix} \begin{pmatrix} \hat{A} \\ \hat{B} \end{pmatrix} \tag{5}$$

if and only if

$$\begin{pmatrix} \left[\hat{H},\hat{A}\right] \\ \left[\hat{H},\hat{B}\right] \end{pmatrix} = \begin{pmatrix} 0 & i\lambda \\ -i\lambda & 0 \end{pmatrix} \begin{pmatrix} \hat{A} \\ \hat{B} \end{pmatrix} \tag{6}$$

For this analysis, it is important to note that the $2 \times 2$ matrix in Eq. (5) is the exponential of the $2 \times 2$ matrix in Eq. (6) multiplied by $-it$ (see SI, section 2):

$$\begin{pmatrix} \cos \lambda t & \sin \lambda t \\ -\sin \lambda t & \cos \lambda t \end{pmatrix} = \exp \begin{pmatrix} 0 & \lambda t \\ -\lambda t & 0 \end{pmatrix} \tag{7}$$

Obviuosly, for this case, it is possible to take a particular parameter $(\lambda)$ of a system of commutator equations and insert it into the template equation (2), which is formulated as a set of two propagation formulae. Then the following questions arise:

1. Is it possible to apply this procedure to more complex cases, such as that of Eq. (4)?

2. To what dimension can we reduce a given problem?

3. Are there any template propagation formulae for the case that a reduction to a 3D or a 4D case is possible?

As a final remark in this subsection, we note that we have performed these considerations in the operator space (Liouville space). This seems advantageous because it is the natural way to treat this topic. We describe states in terms of operators rather than by wave functions, see above. Some properties of the Liouville space that are important for this work are listed in subsection 2.2.

## 2.2 Properties of the Liouville space that are important in this article

In some papers, the space of the wave functions is denoted as Hilbert space in order to have a contrast to the Liouville space. However, this does not seem appropriate since the space of linear operators also satisfies the conditions to be a Hilbert space (Jordan, 2005).

– Definition of the scalar product of two operators $\hat{A}$ und $\hat{B}$: $(\hat{A}, \hat{B}) := \mathrm{Tr} \left( \mathbf{A}^\dagger \cdot \mathbf{B} \right)$ where $\mathbf{A}$ and $\mathbf{B}$ are the matrix representations of $\hat{A}$ and $\hat{B}$ in the wave-function space, and the superscript † means the Hermitian conjugated of the corresponding matrix.

– Two operators $\hat{A}$ and $\hat{B}$ are said to be orthogonal if their scalar product is zero: $\hat{A} \perp \hat{B} \quad \Leftrightarrow \quad \left( \hat{A}, \hat{B} \right) = 0$.

– The Euclidean norm of an operator is defined as the square root of the scalar product of the operator with itself: $\|\hat{A}\| = \sqrt{\left( \hat{A}, \hat{A} \right)} = \sqrt{\mathrm{Tr} \left( \mathbf{A}^\dagger \cdot \mathbf{A} \right)}$. For the sake of brevity, the term "norm" is used throughout this paper to refer to the Euclidean norm. Rules for calculating the norm values for some of the operators used in this article are given in the Appendix. The norm of an operator is invariant with respect to an unitary transformation, but it depends on the relevant space, which is different for different number of spins. Therefore, the table in the appendix contains different norm equations for the same operator but in different spaces;

– Each operator of this space can be expanded into a series of basis operators. In particular, for the density operator $\rho$ we have

$$\hat{\rho} = \rho_1 \hat{u}_1 + ... + \rho_d \hat{u}_d = \mathbf{u}^\mathrm{T} \cdot \boldsymbol{\rho} \tag{8}$$

where $d$ is the dimension of the particular Liouville space, $\boldsymbol{\rho}$ is a column matrix which contains the expansion coefficients $\rho_i$ ($i \in \{1..d\}$) of the density operator, $\mathbf{u}$ is a column matrix the elements of which are the basis operators, and $\mathbf{A}^{\mathrm{T}}$ is the transpose of the matrix $\mathbf{A}$. If all basis operators are pairwise orthogonal, the $\rho_i$ can be calculated as follows

$$\rho_i = (\hat{\rho}, \hat{u}_i) / \|\hat{u}_i\| \qquad (i \in \{1..d\}) \tag{9}$$

– Mappings between operators are described by superoperators; examples:

- Liouville superoperator (or simply "Liouvillian") for forming the commutator with the Hamiltonian: $\hat{\hat{L}} : \hat{A} \mapsto \left[\hat{H}, \hat{A}\right]$, written also as $\hat{\hat{L}}\hat{A} = \left[\hat{H}, \hat{A}\right]$;

- propagation superoperator (or simply "superpropagator") for describing the time evolution: $\hat{\hat{U}} : \hat{\rho}(0) \mapsto \hat{\rho}(t)$, written also as

$$\hat{\rho}(t) = \hat{\hat{U}}\hat{\rho}(0) \tag{10}$$

– The time evolution of the density operator is governed by the Liouville-von Neumann equation. It is formulated in the Liouville space as:

$$\frac{\mathrm{d}}{\mathrm{d}t}\hat{\rho} = -i\hat{\hat{L}}\hat{\rho} \tag{11}$$

If $\hat{\hat{L}}$ does not depend on time, the formal solution is

$$\hat{\rho}(t) = \hat{\hat{U}}(t)\hat{\rho}(0) \quad \text{with} \quad \hat{\hat{U}}(t) = \exp\left(-i\hat{\hat{L}}t\right) \tag{12}$$

Here we find a similarity to the matrix formulation of the POF (Eqs. (5) to (7)). In fact, the coefficient matrix of the system of commutator equations is the transposition of the Liouvillian matrix, and the coefficient matrix of the POF (Eqn. (5)) is the transposition of the superpropagator. This is proved in the SI.

– The norm of the density matrix is time invariant. This can be proven by multiplying the Liouville-vonNeumann equation
(11) scalarly with $\hat{\rho}$:

$$\left(\hat{\rho} \cdot \frac{\mathrm{d}}{\mathrm{d}t}\hat{\rho}\right) = \frac{1}{2}\frac{\mathrm{d}}{\mathrm{d}t}(\hat{\rho} \cdot \hat{\rho}) = -i\left(\hat{\rho} \cdot \hat{\hat{L}}\hat{\rho}\right) = 0 \tag{13}$$

because the scalar product of a hermitian operator ($\hat{\rho}$) with its commutator to another hermitian operator ($\hat{H}$) is zero, see SI.

The Liouville-space formulation of the objective of this article is: Find a subspace of the total Liouville space that contains 
all possible states occurring in the current problem and that has the smallest possible dimension. An analogous procedure will not be possible in the wavefunction space, where the time evolution has to be calculated by $\hat{U}(t)\hat{\rho}(0)\hat{U}^{-1}(t)$. In the following section, commutator equations representing the action of the Liouvillian are established. Their coefficients are needed for further calculation of propagation formulae.

## 2.3 Two forms and two bases for the symbolic description of the time evolution

For practical calculations, Eq. (10) is usually transformed into a matrix equation. In the literature this is usually done in two different forms, both of which can be found for example in Ernst et al. (1987), subsection 2.1.4:

Form 1:

$$\boldsymbol{\rho}(t) = \mathbf{U(t)} \cdot \boldsymbol{\rho(0)} \tag{14}$$

The time evolution is described here in the space of $N$-dimensional column matrices; $\boldsymbol{\rho}(t)$ is the density matrix according to Eq.

(8), i.e. a column containing the coefficients of the expansion of the density operator $\hat{\rho}(t)$ into basis operators. That is, we follow the propagation in the space $\mathbb{C}^N$, the space of $N$-row column matrices with the basis $\{(1,0,...,0)^{\mathrm{T}}, (0,1,...,0)^{\mathrm{T}}, ..., (0,0,...,1)^{\mathrm{T}}\}$. The superpropagator $\hat{\hat{U}}$ is an isomorphism on this space and can be represented by the $N \times N$ matrix $\mathbf{U}$.

Form 2, denoted here as ´propagation formula´:

$$\hat{A}_1 \xrightarrow{\hat{H}t} V_{11}\hat{A}_1 + V_{12}\hat{A}_2 + ... + V_{1N}\hat{A}_N$$

$$\hat{A}_2 \xrightarrow{\hat{H}t} V_{21}\hat{A}_1 + V_{22}\hat{A}_2 + ... + V_{2N}\hat{A}_N$$

$$............$$

$$\hat{A}_N \xrightarrow{\hat{H}t} V_{N1}\hat{A}_1 + V_{N2}\hat{A}_2 + ... + V_{NN}\hat{A}_N \tag{15}$$

This means that the density operator, which is $\hat{A}_1$ at $t = 0$, evolves under the influence of the Hamiltonian $\hat{H}$ during time $t$ into a linear combination of linear independent operators $\hat{A}_1, \hat{A}_2, ..., \hat{A}_N$. Unlike form 1, here we work in the operator space with the basis $\{\hat{A}_1, \hat{A}_2, ..., \hat{A}_N\}$.

Connection between both forms: $\mathbf{U} = \mathbf{V}^{\mathbf{T}}$ if the basis column matrices (form 1) are assigned to the corresponding basis operators of the second basis. For proof, see SI 1.1.

That means, we can rewrite Eq. (15) as

$$\hat{A}_1 \xrightarrow{\hat{H}t} U_{11}\hat{A}_1 + U_{21}\hat{A}_2 + ... + U_{N1}\hat{A}_N$$

$$\hat{A}_2 \xrightarrow{\hat{H}t} U_{12}\hat{A}_1 + U_{22}\hat{A}_2 + ... + U_{N2}\hat{A}_N$$

$$............$$

$$\hat{A}_N \xrightarrow{\hat{H}t} U_{1N}\hat{A}_1 + U_{2N}\hat{A}_2 + ... + U_{NN}\hat{A}_N \tag{16}$$

The set of all $\mathbf{U}$ is the dual space of the set of all $\mathbf{V}$. Similarly, the coefficient matrix of the commutator equations is the transposed Liouvillian matrix, see SI 1.1.

# 3 Procedure for finding propagation formulae

## 3.1 Requirements for a suitable subspace

To be sure that a given subspace $\mathbb{S}$ contains the whole evolution of a spin system, we have to check that the action of the propagator $\hat{\hat{U}}$ on each operator $\hat{A}$ of this subspace also results in an element of this subspace:

$$\forall \hat{A} \in \mathbb{S} \quad : \quad \hat{\hat{U}} \, \hat{A} \in \mathbb{S} \tag{17}$$

A subspace with this property is said to be propagator invariant (Jordan, 2005). This is equivalent to the Liouvillian invariance of the subspace, i.e.

$$\forall \hat{A} \in \mathbb{S} \quad : \quad \hat{\hat{L}} \, \hat{A} \in \mathbb{S} \tag{18}$$

because the propagator is the sum of repeated $\hat{\hat{L}}$ actions:

$$\hat{\hat{U}} \hat{A} = \exp\left(-i\hat{\hat{L}}t\right) \hat{A} = \sum_{n=0}^{\infty} \frac{(-it)^n}{n!} \hat{\hat{L}}^n \, \hat{A} \quad \in \mathbb{S} \tag{19}$$

A sufficient condition for a subspace spanned by $\hat{u}_1 ... \hat{u}_N$ (Eq. (8)) to be Liouvillian invariant is the requirement that the action of the Liouvillian on any of the $\hat{u}_i$ also results in an element of that subspace, i.e. $\hat{\hat{L}} \hat{u}_i \in \mathbb{S} \quad \forall i \in \{1..N\}$. A Liouvillian invariant subspace which is of interest here should at least contain the initial density operator. Furthermore, all multiple actions of the Liouvillian on the density operator must also result in elements of that subspace: $\hat{\hat{L}}^n \, \hat{\rho}(0) \in \mathbb{S} \quad \forall n \in \mathbb{N}$.

In principle we can construct such a subspace as the set of all linear combinations of $\hat{\rho}(0), \hat{\hat{L}}\hat{\rho}(0), \hat{\hat{L}}^2\hat{\rho}(0), ..., \hat{\hat{L}}^N\hat{\rho}(0)$, where $N$ is the largest number for which this operator set is linearly independent. This means that $\hat{\hat{L}}^{N+1}\hat{\rho}(0)$ can be represented as a linear combination of all lower powers. Then all further applications of $\hat{\hat{L}}$ lead to operators which are also linearly dependent. This procedure is reminiscent of the formation of Krylov subspaces in matrix spaces (Watkins, 2007). For further considerations in this article, in particular the calculation of matrices, it is convenient to have basis operators that are pairwise orthogonal. This can be achieved by combining the Krylov-like procedure with the Gram-Schmidt orthogonalization, which is analogous to the Arnoldi procedure for matrix spaces (Watkins, 2007). The details of the procedure are presented in the next subsection.

## 3.2 First step: Creating a closed system of commutator equations

According to the result of the previous subsection, the application of the Liouvillian to the initial density operator is repeated as long as the result contains a component that is linearly independent of all previous results. The action of the Liouvillian consists in forming the commutator of the Hamiltonian with the considered operator. Consequently, the search for this Liouvillian-invariant minimum subspace is performed by repeatedly calculating the commutator of the Hamiltonian with the operator representing the initial state of the spin system, as shown in detail below.

Let us assume that the system under consideration is characterized by the Hamiltonian $\hat{H}$ and the initial state density operator by $\hat{A}_1$.

Evaluation of the first commutator: $\left[\hat{H},\hat{A}_1\right]$

If the result is zero, the state of the spin system is constant in time; see SI, examples 0D-1 and 0D-2. In the case that this commutator does not vanish, it can be decomposed into a term which is proportional to $\hat{A}_1$ and another term $\hat{A}_{12}$ which is orthogonal to the first one:

$$\left[\hat{H},\hat{A}_1\right] = \lambda_{11}\hat{A}_1 + \hat{A}_{12} \qquad \text{with} \quad \hat{A}_{12} \perp \hat{A}_1 \tag{20}$$

The commutator of two Hermitian operators is always orthogonal to both, as proved in the SI. This means that for a Hermitian $\hat{A}_1$, $\left[\hat{H},\hat{A}_1\right]$ cannot have a component that contains $\hat{A}_1$ itself. This is different for non-Hermitian operators, see subsection 4.1.

We replace $\hat{A}_{12} \to \lambda_{12}\hat{A}_2$, choosing the scalar $\lambda_{12}$ so that $\hat{A}_2$ has the same norm as $\hat{A}_1$. The corresponding table in the appendix can be used to determine the norms of the operators.

Evaluation of the second commutator: $\left[\hat{H},\hat{A}_2\right]$

It will be decomposed as

$$\left[\hat{H},\hat{A}_2\right] = \lambda_{21}\hat{A}_1 + \lambda_{22}\hat{A}_2 + \lambda_{23}\hat{A}_3 \tag{21}$$

with the condition that $\hat{A}_3$ is orthogonal to both $\hat{A}_1$ and $\hat{A}_2$. The coefficient $\lambda_{23}$ is chosen so that $\hat{A}_3$ has the same norm as $\hat{A}_1$ and $\hat{A}_2$.

If $\hat{A}_3 = 0$, i.e. $\left[\hat{H},\hat{A}_2\right]$ is a linear combination of $\hat{A}_1$, the procedure is finished. Then we have a system of commutator equations like Eq. (3), i.e. the usual POF.

Evaluation of the $n$-th commutator: $\left[\hat{H},\hat{A}_n\right]$

Similarly, it will be expanded into a series of operators known from the previous commutator evaluations and a remainder:

$$\left[\hat{H},\hat{A}_n\right] = \lambda_{n1}\hat{A}_1 + ... + \lambda_{n,n}\hat{A}_n + \lambda_{n,n+1}\hat{A}_{n+1} \tag{22}$$

with the condition that $\hat{A}_{n+1}$ is orthogonal to all of the $\hat{A}_k$ with $k \in 1..n$. Again, the coefficient $\lambda_{n,n+1}$ is chosen so that $\hat{A}_{n+1}$ has the same norm as the other operators of this set.

End of the procedure: If for a certain $n = N$ the commutator is a linear combination of the previously determined $\hat{A}_k$ without any remainder, the iteration is finished. The set of pairwise orthogonal operators $\{\hat{A}_1..\hat{A}_N\}$ spans a Liouvillian-invariant subspace of the entire Liouville space. Its dimension is $N$.

Possible modification of the procedure: The remainder of any commutator evaluation can be written as the sum of two or more operators, all of which must be orthogonal to the other operators. In some cases, this can simplify the coefficient matrix.

 ## 3.3 Second step: Liouvillian matrix

The result of the whole procedure is a system of equations like the following:

$$
\begin{pmatrix}
\left[\hat{H}, \hat{A}_1\right] \\
\left[\hat{H}, \hat{A}_2\right] \\
\left[\hat{H}, \hat{A}_3\right] \\
... \\
\left[\hat{H}, \hat{A}_{N-2}\right] \\
\left[\hat{H}, \hat{A}_{N-1}\right] \\
\left[\hat{H}, \hat{A}_N\right]
\end{pmatrix}
=
\begin{pmatrix}
\lambda_{11} & \lambda_{12} & 0 & ... & 0 & 0 & 0 \\
\lambda_{21} & \lambda_{22} & \lambda_{23} & ... & 0 & 0 & 0 \\
0 & \lambda_{32} & \lambda_{33} & ... & 0 & 0 & 0 \\
... & ... & ... & ... & ... & ... & ... \\
0 & 0 & 0 & ... & \lambda_{N-2,N-2} & \lambda_{N-2,N-1} & 0 \\
0 & 0 & 0 & ... & \lambda_{N-1,N-2} & \lambda_{N-1,N-1} & \lambda_{N-1,N} \\
0 & 0 & 0 & ... & 0 & \lambda_{N,N-1} & \lambda_{N,N}
\end{pmatrix}
\begin{pmatrix}
\hat{A}_1 \\
\hat{A}_2 \\
\hat{A}_3 \\
... \\
\hat{A}_{N-2} \\
\hat{A}_{N-1} \\
\hat{A}_N
\end{pmatrix}
\tag{23}
$$

As shown in SI 1.1, the coefficient matrix is the transposition of the Liouvillian matrix. This system of equations implies that the action of the Liouvillian on any $\hat{A}_i$ ($i \in \{1..N\}$) leads to a linear combination of the $\hat{A}_i$: In other words, the subspace spanned by the operators $\hat{A}_1 ... \hat{A}_n$ is both Liouvillian invariant and propagator invariant. The density operator, once being located in this subspace, will not leave it as long as the interaction does not vary. This explains why the POF can be successfully applied as a 2D problem, even if the complete Liouville space has a much higher dimension.

The zeros in the upper triangle of the coefficient matrix result from the above: If the remainder of the $n$-th commutator is identified with only one new operator, then $\lambda_{n,n+1}$ and $\hat{A}_{n+1}$ are determined. $\lambda_{n,n+2},...$ and $\hat{A}_{n+2},...$ are still unknown at this step; the matrix elements to the right of them remain zero. In the case of the modified procedure mentioned above, the matrix structure in Eq. (23) changes. Then $\lambda_{n,n+2}...$ can also be nonzero. Due to the Hermiticity of the Liouvillian, i.e. $L_{ij} = L_{ji}^*$, the lower triangle matrix has the same pattern of zeros as the upper one. The coefficient matrix has a band structure for the unmodified version of the procedure.

When the basis operators are Hermitian, all elements are purely imaginary. This means that all elements of the main diagonal must be zero. However, if the basis contains non-Hermitian operators, the main diagonal will contain non-zero elements, but they must be real numbers.

## 3.4 Third step: Estimation of propagator matrix and propagation rules

In principle, the propagator matrix can be obtained by evaluating the matrix exponential according to Eq. (12):

$$
\mathbf{U} = \exp(-i\mathbf{L}t) =
\begin{pmatrix}
U_{11} & U_{12} & ... & U_{1N} \\
U_{21} & U_{22} & ... & U_{2N} \\
... & ... & ... & ... \\
U_{N1} & U_{N2} & ... & U_{NN}
\end{pmatrix}
\tag{24}
$$

**L** is a pure imaginary matrix for an hermitian operator basis. In this case, **U** contains only real elements and is orthogonal. This means that the norms of all rows and all columns of this matrix are unity:

$$\sum_{i=1}^{N} U_{ij}^2 = \sum_{j=1}^{N} U_{ij}^2 = 1 \tag{25}$$

The propagation rules can be obtained from the elements of the transposed propagator matrix (proof see SI 1.1):

$$\hat{A}_1 \xrightarrow{\hat{H}t} \hat{A}_1 U_{11} + \hat{A}_2 U_{21} + ... + \hat{A}_N U_{N1}$$
$$\hat{A}_2 \xrightarrow{\hat{H}t} \hat{A}_1 U_{12} + \hat{A}_2 U_{22} + ... + \hat{A}_N U_{N2}$$
$$... \tag{26}$$

However, it is not necessary to recompute the matrix exponential for each new situation. Instead, for low-dimensional subspaces, the generic propagation formulae shown in the next section can be used as a template. Here the elements of the Liouvillian matrix have to be inserted directly, without the need to perform the matrix exponentialization. This was done above for the POF example, where the constant $\lambda$ resulting from the commutator equations could be used directly as the oscillation frequency.

## 4    Special cases

The situations in the examples shown below are characterized by different initial states and different Hamiltonians. The Hamiltonians are listed in Appendix B.

A detailed analytical consideration for each example can be found in the SI.

### 4.1    Reduction to a one-dimensional subspace

In this case, the commutator of the Hamiltonian with the density operator at $t = 0$, i.e. $\hat{A}_1$, is proportional to $\hat{A}_1$ itself:

$$\left[\hat{H}, \hat{A}_1\right] = \lambda \hat{A}_1 \tag{27}$$

and occurs only if $\hat{A}_1$ is non-Hermitian, e.g. $\hat{A}_1 = \hat{I}_+ \equiv \hat{I}_x + i\hat{I}_y$ as used for the characterization of the complex FID, see Abragam (1961). Then the coefficient matrix consists of only one scalar $\lambda$. The propagator is also scalar:

$$\mathbf{U}_{1D} = e^{-i\lambda t} \tag{28}$$

and the corresponding propagation rule is

$$\hat{A}_1 \xrightarrow{\hat{H}t} \hat{A}_1 e^{-i\lambda t} \tag{29}$$

Example: Rotating frame, resonance offset $\Delta\omega$, complex transversal magnetization represented by $\hat{I}_+$:

$$\hat{I}_+ \xrightarrow{-\Delta\omega\hat{I}_z t} \hat{I}_+ e^{-i\Delta\omega t} \tag{30}$$

## 4.2 Case of reduction to a 2D subspace

The corresponding equations are like Eqns. (3) and (6) and belong to the POF. The Liouvillian matrix and the propagator matrix are the transpositions of the matrices appearing in Eqs. (5) and (6):

$$\mathbf{L}_{2D} = \begin{pmatrix} 0 & -i\lambda \\ i\lambda & 0 \end{pmatrix} ; \qquad \mathbf{U}_{2D} = \exp(-i\mathbf{L}_{2D}t) = \begin{pmatrix} \cos\lambda t & -\sin\lambda t \\ \sin\lambda t & \cos\lambda t \end{pmatrix} \tag{31}$$

The propagation formulae are given by Eqn. (2). It describes an oscillatory behavior between the initial state and another state described by the commutator of the Hamiltonian with the operator corresponding to the density operator at the beginning.

In addition to the cases known from numerous POF applications, there are other situations that can be described as time evolution in a 2D subspace. All of them are well known; they are listed here for sake of completeness:

- FID of an ensemble of isolated pairs of equal spins ($I_1$, $I_2$) after a $\pi/2$ pulse; homonuclear dipolar interaction within the spin pairs; we observe the transversal magnetization represented by the operator sum $\hat{I}_{1x} + \hat{I}_{2x}$:

$$\hat{I}_{1x} + \hat{I}_{2x} \quad \xrightarrow{\hat{H}_{II}\,t} \quad \left(\hat{I}_{1x} + \hat{I}_{2x}\right)\cos\frac{3}{2}D_{II}t - 2\left(\hat{I}_{1z}\hat{I}_{2y} + \hat{I}_{1y}\hat{I}_{2z}\right)\sin\frac{3}{2}D_{II}t \tag{32}$$

- FID of an ensemble of isolated pairs of unlike spins ($I$, $S$) after a $\pi/2$ pulse in the $I$ channel; heteronuclear dipolar interaction within the spin pairs; we observe the transversal $I$ magnetization represented by the operator $\hat{I}_x$:

$$\hat{I}_x \quad \xrightarrow{\hat{H}_{IS}\,t} \quad \hat{I}_x \cos D_{IS}t - 2\hat{I}_y\hat{S}_z \sin D_{IS}t \tag{33}$$

- FID of an ensemble of spins $I = 1$ (e.g., $^2H$ or $^{14}N$) under quadrupolar interaction; we follow again the transversal magnetization:

$$\hat{I}_x \quad \xrightarrow{\hat{H}_Q\,t} \quad \hat{I}_x \cos\omega_Q t + \left(\hat{I}_z\hat{I}_y + \hat{I}_y\hat{I}_z\right)\sin\omega_Q t \tag{34}$$

- Ensemble of pairs of homonuclearly coupled equal spins ($I_1$, $I_2$) with spin quantum number 1/2, where initially the spin 1 is oriented parallel to $\mathbf{B}_0$ and the spin 2 antiparallel to that. We follow the difference $z$ magnetization which will be represented by $\hat{I}_{1z} - \hat{I}_{2z}$:

$$\hat{I}_{1z} - \hat{I}_{2z} \quad \xrightarrow{\hat{H}_{II}\,t} \quad \left(\hat{I}_{1z} - \hat{I}_{2z}\right)\cos D_{II}t + 2\left(\hat{I}_{1x}\hat{I}_{2y} - \hat{I}_{1y}\hat{I}_{2x}\right)\sin D_{II}t \tag{35}$$

- Cross polarization within pairs of antiparallel unlike spins ($I$, $S$): Both spins are locked in resonant rf fields with equal nutation frequencies $\omega_{1I} = \omega_{1S} \gg D_{IS}$ (Hartmann-Hahn condition). The Hamiltonian and the state operators are given in the doubly-rotating frame following Hartmann and Hahn (1962) where the $z$ direction is along the rf irradiation. If initially the $S$ spins are oriented parallel to the locking field and the $I$ spins are antiparallel to that, the time evolution can be described by following $\hat{S}_z - \hat{I}_z$:

$$\hat{S}_z - \hat{I}_z \quad \xrightarrow{\hat{H}_{HH}\,t} \quad \left(\hat{S}_z - \hat{I}_z\right)\cos D_{IS}t + 2\left(\hat{I}_x\hat{S}_y - \hat{I}_y\hat{S}_x\right)\sin D_{IS}t \tag{36}$$

The last equation describes the behavior of the difference of $I$ and $S$ polarization, not the individual polarizations themselves. The time evolution of the latter requires at least a 3D approach, see below.

The oscillation takes place in the first three examples between observable transversal magnetization and antiphase states, in the last two examples between longitudinal difference magnetization and zero and double quantum coherences. These examples show an effect of the dimension reduction: To obtain a 2D problem, the operators characterizing the states of the spin system have a more complicated structure than in the simple cases above. For example, it would be possible, to consider the right side of Eq. (32) as a linear combination of the four states $\hat{I}_{1x}, \hat{I}_{2x}, 2\hat{I}_{1z}\hat{I}_{2y}$, and $2\hat{I}_{1y}\hat{I}_{2z}$ if a more illustrative notation is desired.

## 4.3 Case of reduction to a 3D subspace

### 4.3.1 Generic notation

Here we are dealing with those cases where the procedure described above reaches the cancellation condition after three commutator equations of the kind

$$
\begin{aligned}
\left[\hat{H}, \hat{A}\right] &= ia\hat{B} \\
\left[\hat{H}, \hat{B}\right] &= -ia\hat{A} + ib\hat{C} \\
\left[\hat{H}, \hat{C}\right] &= -ib\hat{B}
\end{aligned}
\tag{37}
$$

where $a, b \in \mathbb{R}$. In step 2, we determine the Liouvillian matrix as transposed coefficient matrix of Eq. (37):

$$
\mathbf{L}_{3D} = \begin{pmatrix} 0 & -ia & 0 \\ ia & 0 & -ib \\ 0 & ib & 0 \end{pmatrix}
\tag{38}
$$

and from that the matrix of the superpropagator as matrix exponential corresponding to Eq. (24):

$$
\mathbf{U}_{3D} = \exp(-i\mathbf{L}_{3D}t) = \frac{1}{q^2} \begin{pmatrix} b^2 + a^2\cos qt & -aq\sin qt & ab(1-\cos qt) \\ aq\sin qt & q^2\cos qt & -bq\sin qt \\ ab(1-\cos qt) & bq\sin qt & a^2 + b^2\cos qt \end{pmatrix}
\tag{39}
$$

with $q^2 := a^2 + b^2$. The orthogonality of $\mathbf{U}_{3D}$, i.e. the validity of Eqn. (25), can be verified immediately.

In step 3, according to Eq. (26), the following propagation rules are obtained from the columns of $\mathbf{U}_{3D}$:

$$
\hat{A} \xrightarrow{\hat{H}\,t} \hat{A} \cdot \frac{b^2 + a^2\cos qt}{q^2} \qquad +\hat{B} \cdot \frac{a}{q}\sin qt \qquad +\hat{C} \cdot \frac{ab}{q^2}(1-\cos qt)
\tag{40}
$$

$$
\hat{B} \xrightarrow{\hat{H}\,t} \hat{B} \cdot \cos qt \qquad +\hat{C} \cdot \frac{b}{q}\sin qt \qquad -\hat{A} \cdot \frac{a}{q}\sin qt
\tag{41}
$$

$$
\hat{C} \xrightarrow{\hat{H}\,t} \hat{C} \cdot \frac{a^2 + b^2\cos qt}{q^2} \qquad +\hat{A} \cdot \frac{ab}{q^2}(1-\cos qt) \qquad -\hat{B} \cdot \frac{b}{q}\sin qt
\tag{42}
$$

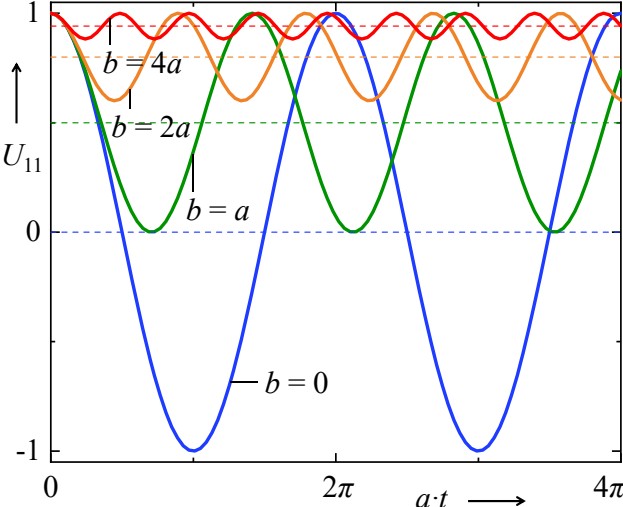

**Figure 1.** Time evolution of the prefactor of $\hat{A}$ in Eqn. (40) for different ratios $b/a$. The particular case $b = 0$ leads to the 2D case; therefore, the corresponding curve is a pure oscillation around zero. With increasing $b$, the average levels of the related oscillations (dashed horizontal lines) increase whereas the corresponding amplitudes decrease.

This can be seen as an extension of the POF to three-dimensional problems.

If all basis operators are Hermitian, one of the eigenvalues of $\mathbf{L}_{3D}$ is zero because of $\det \mathbf{L}_{3D} = 0$. Therefore, the solution of the Liouville-von Neumann equation may contain a non-zero constant beyond the oscillating terms. The propagator-matrix element $(U_{3D})_{11}$ contains the time evolution of the initial state $\hat{A}$. The constant term $b^2/(a^2 + b^2)$ shows that the oscillations do not take place around zero as in the 2D case, but around another level. Moreover, its amplitude is reduced to $a^2/(a^2 + b^2)$, while the frequency increases the smaller the amplitude is, see Fig. 1.

There is one important special case: $a = b$. Here the propagation formulae are simplified to:

$$\hat{A} \xrightarrow{\hat{H}\,t} \hat{A} \cdot \cos^2 \frac{qt}{2} \qquad\qquad + \hat{B} \cdot \frac{1}{\sqrt{2}} \sin qt \qquad\qquad + \hat{C} \cdot \sin^2 \frac{qt}{2} \tag{43}$$

$$\hat{B} \xrightarrow{\hat{H}\,t} \hat{B} \cdot \cos qt \qquad\qquad + \hat{C} \cdot \frac{1}{\sqrt{2}} \sin qt \qquad\qquad - \hat{A} \cdot \frac{1}{\sqrt{2}} \sin qt \tag{44}$$

$$\hat{C} \xrightarrow{\hat{H}\,t} \hat{C} \cdot \cos^2 \frac{qt}{2} \qquad\qquad + \hat{A} \cdot \sin^2 \frac{qt}{2} \qquad\qquad - \hat{B} \cdot \frac{1}{\sqrt{2}} \sin qt \tag{45}$$

This is applied in the description of cross polarization and polarization transfer, see the corresponding examples below.

### 4.3.2 Group 1 of experiments leading to 3D subspaces: Magnetization initially aligned parallel to $B_0$

– Off-resonance nutation: rf irradiation with strength $\omega_1$ on an ensemble of isolated spins under resonance offset $\Delta\omega$:

$$\hat{I}_z \xrightarrow{(\hat{H}_{Ix} + \hat{H}_\Delta)\,t} \hat{I}_z \cdot \frac{\Delta\omega^2 + \omega_1^2 \cos qt}{q^2} + \hat{I}_y \cdot \frac{\omega_1}{q} \sin qt + \hat{I}_x \cdot \frac{\omega_1 \Delta\omega}{q^2}(1 - \cos qt) \tag{46}$$

with $q^2 = \omega_1^2 + \Delta\omega^2$.

– Nutation and dipolar interaction: The observed spin $I$, which is heteronuclearly coupled to the spin $S$, is additionally irradiated with rf of the strength $\omega_{1I}$:

$$\hat{I}_z \xrightarrow{\left(\hat{H}_{Ix}+\hat{H}_{IS}\right)t} \hat{I}_z \cdot \frac{D^2 + \omega_{1I}^2 \cos qt}{q^2} + \hat{I}_y \cdot \frac{\omega_{1I}}{q}\sin qt + 2\hat{I}_x\hat{S}_z \cdot \frac{\omega_{1I}D}{q^2}(1-\cos qt) \tag{47}$$

with $q^2 = \omega_{1I}^2 + D^2$.

(The case of *homonuclear* interaction under rf irradiation leads to a 4D problem, see below.)

The similarity of the two equations (46) and (47) is obvious. For $\Delta\omega \longrightarrow 0$ and $D_{IS} \longrightarrow 0$, respectively, they merge into an equation describing a rotation in the $yz$ plane. Both equations reflect the well-known fact that a total inversion of the magnetization with a single rectangular pulse is only possible if the offset or the coupling is zero. In addition, coupling and resonance offset change both the pulse duration $\tau_{\pi/2}$ required to reach maximum $y$ magnetization and the pulse duration $\tau_\pi$ required to reach zero $y$ magnetization to shorter times:

$$\tau_{\pi/2} = \frac{\pi}{2\sqrt{\omega_{1I}^2 + C^2}}\;; \qquad \tau_\pi = \frac{\pi}{\sqrt{\omega_{1I}^2 + C^2}} \tag{48}$$

with $C = \Delta\omega$ for the off-resonance nutation (Eqn. (46)) and $C = D_{IS}$ for the nutation under heteronuclear dipolar interaction.

### 4.3.3 Group 2 of experiments leading to 3D subspaces: FID under both rf irradiation and dipolar interaction

– Decoupling experiment: Ensemble of spin pairs $\{IS\}$; heteronuclear dipolar interaction between $I$ and $S$, rf irradiation on $S$ channel with finite rf power of strength $\omega_{1S}$; spins $I$ are observed:

$$\hat{I}_x \xrightarrow{\left(\hat{H}_{Sx}+\hat{H}_{IS}\right)t} \hat{I}_x \cdot \frac{\omega_{1S}^2 + D_{IS}^2 \cos qt}{q^2} - 2\hat{I}_y\hat{S}_z \cdot \frac{D_{IS}}{q}\sin qt - 2\hat{I}_x\hat{S}_z \cdot \frac{\omega_{1S}D_{IS}}{q^2}(1-\cos qt) \tag{49}$$

with $q^2 = \omega_{1S}^2 + D_{IS}^2$. For obtaining the corresponding equation for the $J$ coupling, replace $D_{IS}$ with $-\pi J$. This equation describes a *partial* exchange of polarization between $x$ magnetization and two antiphase states.

The decoupling effect is explained as follows: The observable part of the density operator - the prefactor of $\hat{I}_x$ - contains a constant part $\omega_{1S}^2/\left(\omega_{1S}^2 + D_{IS}^2\right)$ and an oscillating part with the amplitude $D_{IS}^2/\left(\omega_{1S}^2 + D_{IS}^2\right)$. Such oscillations were observed for instance in DIPSHIFT experiments (Kurz et al., 2013). Powder averaging leads to a rather fast decay of the oscillation which gives a broad line (Pake doublet) after Fourier transformation, while the former gives a $\delta$ line. With increasing rf strength $\omega_{1S}$, the prefactor of the broad peak decreases to zero for infinite rf power, while that of the $\delta$ line increases. The constant component is subject to relaxation damping and chemical-shift-caused oscillation on a longer time scale and produces a more or less narrow line.

– On-resonance spin locking and heteronuclear dipolar coupling:

$$\hat{I}_x \xrightarrow{\left(\hat{H}_{Ix}+\hat{H}_{IS}\right)t} \hat{I}_x \cdot \frac{\omega_{1I}^2 + D_{IS}^2 \cos qt}{q^2} - 2\hat{I}_y\hat{S}_z \cdot \frac{D_{IS}}{q}\sin qt + 2\hat{I}_z\hat{S}_z \cdot \frac{\omega_{1I}D_{IS}}{q^2}(1-\cos qt) \tag{50}$$

with $q^2 = \omega_{1I}^2 + D_{IS}^2$. This oscillation frequency is indeed equal to that of the corresponding nutation experiment, see equation (47).

– On-resonance spin locking and homonuclear dipolar coupling:

$$\hat{I}_{1x} + \hat{I}_{2x} \quad \xrightarrow{\left(\hat{H}_{Ix} + \hat{H}_{II}\right)t} \quad \left(\hat{I}_{1x} + \hat{I}_{2x}\right) \cdot \frac{4\omega_{1I}^2 + \frac{9}{4}D_{II}^2 \cos qt}{q^2} - 2\left(\hat{I}_{1y}\hat{I}_{2z} + \hat{I}_{1z}\hat{I}_{2y}\right) \cdot \frac{3D_{II}}{2q}\sin qt$$

$$+ 2\left(\hat{I}_{1z}\hat{I}_{2z} - \hat{I}_{1y}\hat{I}_{2y}\right) \cdot \frac{3\omega_{1I}D_{II}}{q^2}(1 - \cos qt) \tag{51}$$

with $q^2 \rightarrow 4\omega_{1I}^2 + \frac{9}{4}D_{IS}^2$.

Comments on the spinlock examples (Eqns. (50) and (51)):

– Analogous to the other 3D examples, the oscillation takes place around a level that increases with $\omega_{1I}$. The latter corresponds to the spin-locked part of the transversal magnetization. At the same time, the amplitude of the oscillation is reduced.

– These oscillations are observed at the beginning of spin-lock experiments (Krushelnitsky et al., 2018, 2023) and have been described theoretically by Garroway (1979) and Mcarthur et al. (1969). Due to their orientation dependence, they decay quite rapidly in a powder sample, but can be refocused in MAS experiments.

– The propagation formula (50) is related to the same Hamiltonian as formula (47). As a consequence, the oscillation frequencies are the same. However, the different initial states lead to different subspaces and thus to different propagation formulae.

### 4.3.4 Group 3 of experiments leading to 3D subspaces: Polarization transfer

The rf field strengths are assumed to be much larger than the corresponding coupling frequencies. This situation is very similar to the polarization transfer treated as 2D case. The difference lies in the initial states. Instead of the antiparallel orientation above, here one spin of the pairs is polarized and the other is not. The initial states are now only described by $\hat{I}_{1z}$ and $\hat{S}_z$, resp. Since these operators are not elements of the 2D subspaces of the polarization difference examples above, the motion now takes place in other subspaces, which turn out to be three-dimensional.

– Equal spins:

$$\hat{I}_{1z} \quad \xrightarrow{\hat{H}_{II}\,t} \quad \hat{I}_{1z} \cdot \cos^2\left(\frac{D_{II}t}{2}\right) + \left(\hat{I}_{1x}\hat{I}_{2y} + \hat{I}_{1y}\hat{I}_{2x}\right) \cdot \sin D_{II}t + \hat{I}_{2z} \cdot \sin^2\left(\frac{D_{II}t}{2}\right) \tag{52}$$

– Pair of unequal spins $I, S$ under Hartmann-Hahn condition in the doubly-rotating frame:

$$\hat{S}_z \quad \xrightarrow{\hat{H}_{HH}\,t} \quad \hat{S}_z \cdot \cos^2\left(\frac{D_{IS}t}{2}\right) + \left(\hat{I}_x\hat{S}_y + \hat{I}_y\hat{S}_x\right) \cdot \sin D_{IS}t + \hat{I}_z \cdot \sin^2\left(\frac{D_{IS}t}{2}\right) \tag{53}$$

Muller et al. (1974) were the first to experimentally demonstrate this oscillatory exchange of polarisation during cross-polarisation.

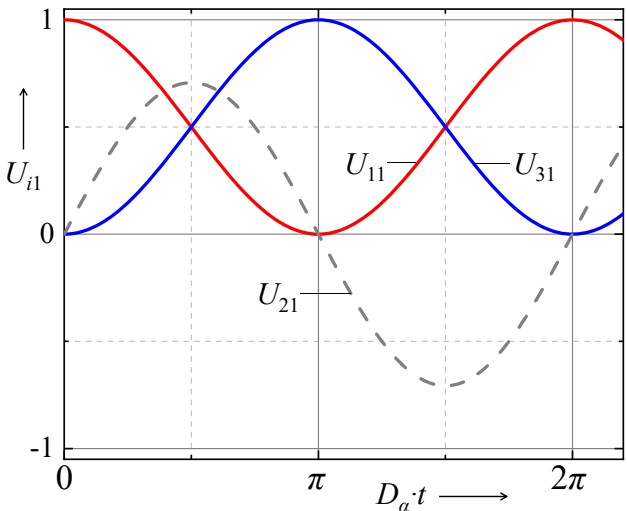

**Figure 2.** Time evolution of the three prefactors in the propagation rules Eqns. (52) and (53).

– Depolarization of $I$ spins in an ensemble of spin triples $\{I, S_1, S_2\}$ under Hartmann-Hahn condition in the doubly rotating frame; coupling frequencies for the $I - S_1$ and the $I - S_2$ interactions: $D_1$ and $D_2$, respectively. The interaction between $S_1$ and $S_2$ is assumed to be zero. This can be realized experimentally by irradiating the S spins with a resonance offset which is $1/\sqrt{2}$ times the rf strength, known as Lee-Goldburg condition, see Lee and Goldburg (1965).

$$
\hat{I}_z \quad \xrightarrow{\hat{H}_{\text{HH2}} \cdot t} \quad \hat{I}_z \cdot \cos^2\left(\frac{\sqrt{D_1^2 + D_2^2}}{2} t\right)
$$

$$
- \frac{D_1\left(\hat{I}_x \hat{S}_{1y} - \hat{I}_y \hat{S}_{1x}\right) + D_2\left(\hat{I}_x \hat{S}_{2y} - \hat{I}_y \hat{S}_{2x}\right)}{\sqrt{D_1^2 + D_2^2}} \sin \sqrt{D_1^2 + D_2^2}\, t
$$

$$
+ \frac{D_1^2 \hat{S}_{1z} + D_2^2 \hat{S}_{2z} - 4 D_1 D_2 \, \hat{I}_z \left(\hat{S}_{1x} \hat{S}_{2x} + \hat{S}_{1y} \hat{S}_{2y}\right)}{D_1^2 + D_2^2} \sin^2\left(\frac{\sqrt{D_1^2 + D_2^2}}{2} t\right) \tag{54}
$$

Comment on the third item: The oscillation frequency is the geometric sum of the individual frequencies. The oscillation takes place between the initial state and a mixture of observable and unobservable states.

Note the first two propagation rules of the third group: Additionally to the orthogonality relations (Eqn. (25)), the linear sum of the prefactors of the first and the third terms is 1. Both terms represent $z$ magnetization. This condition reflects that the sum of the $z$ polarizations is constant for CP as well as for PT. This is supported by the fact that $\hat{I}_{1z} + \hat{I}_{2z}$ commutes with $\hat{H}_{\text{II}}$ and $\hat{I}_z + \hat{S}_z$ commutes with $\hat{H}_{\text{HH}}$. Fig. 2 shows the time evolution of the three prefactors in these propagation rules.

### 4.3.5 Group 4 of 3D examples: Cross polarization, finite rf power, possible deviation from HH condition

– Considering the polarization difference in the rotating frame:

$$\hat{S}_x - \hat{I}_x \xrightarrow{\ \left(\hat{H}_{Ix}+\hat{H}_{Sx}+\hat{H}_{IS}\right)t\ } \left(\hat{S}_x - \hat{I}_x\right) \cdot \frac{\omega_\Delta^2 + D_{IS}^2 \cos q_\Delta t}{q_\Delta^2}$$

$$-2\left(\hat{I}_z\hat{S}_y - \hat{I}_y\hat{S}_z\right) \cdot \frac{D_{IS}}{q_\Delta} \sin q_\Delta t$$

$$+2\left(\hat{I}_z\hat{S}_z + \hat{I}_y\hat{S}_y\right) \cdot \frac{\omega_\Delta D_{IS}}{q_\Delta^2} \sin^2 q_\Delta t \tag{55}$$

with $\omega_\Delta = \omega_{1S} - \omega_{1I}$ and $q_\Delta^2 = D_{IS}^2 + \omega_\Delta^2$.

Comments:

– The CP oscillation frequency is no longer $D_{IS}$ as calculated for infinite rf power (see above), but $\sqrt{D_{IS}^2 + (\omega_{1I} - \omega_{1S})^2}$ which increases with the difference of the two rf field strengths.

– Moreover, only the relative part $D_{IS}^2/q_\Delta^2$ of the total magnetization participates in the oscillation, i.e. the greater the deviation from the Hartmann-Hahn condition, the lower the maximum transmitted polarization.

– Considering the polarization sum:

$$\hat{S}_x + \hat{I}_x \xrightarrow{\ \left(\hat{H}_{Ix}+\hat{H}_{Sx}+\hat{H}_{IS}\right)t\ } \left(\hat{S}_x + \hat{I}_x\right) \cdot \frac{4\omega_\emptyset^2 + D_{IS}^2 \cos q_\emptyset t}{q_\emptyset^2}$$

$$-2\left(\hat{I}_z\hat{S}_y + \hat{I}_y\hat{S}_z\right) \cdot \frac{D_{IS}}{q_\emptyset} \sin q_\emptyset t$$

$$+2\left(\hat{I}_z\hat{S}_z - \hat{I}_y\hat{S}_y\right) \cdot \frac{2\omega_\emptyset D_{IS}}{q_\emptyset^2} \sin^2 q_\emptyset t \tag{56}$$

with $\omega_\emptyset = \left(\omega_{1S} + \omega_{1I}\right)/2$ and $q_\emptyset^2 = D_{IS}^2 + 4\omega_\emptyset^2$.

Contrary to the relations obtained for infinite $\omega_1$, the sum of both polarizations oscillates. The amplitude decreases with increasing rf power. This phenomenon is analogous to what happens with spin-lock and decoupling, see the corresponding examples above.

## 4.4 Case of reduction to a 4D subspace

### 4.4.1 Generic notation

This group of situations can be described by commutator relations of the kind

$$
\begin{aligned}
\left[\hat{H}, \hat{A}\right] &= ia\hat{B} \\
\left[\hat{H}, \hat{B}\right] &= -ia\hat{A} + ib\hat{C} \\
\left[\hat{H}, \hat{C}\right] &= -ib\hat{B} \mp ia\hat{D} \\
\left[\hat{H}, \hat{D}\right] &= \pm ia\hat{C}
\end{aligned}
\tag{57}
$$

where $\hat{A}$, $\hat{B}$, $\hat{C}$ and $\hat{D}$ are pairwise orthogonal operators with equal norms. According to step 2, we obtain the Liouvillian matrix from these rules as transposed coefficient matrix:

$$
\mathbf{L}_{3D} =
\begin{pmatrix}
0 & -ia & 0 & 0 \\
ia & 0 & -ib & 0 \\
0 & ib & 0 & \pm ia \\
0 & 0 & \mp ia & 0
\end{pmatrix}
\tag{58}
$$

and the superpropagator matrix as

$$
\mathbf{U}_{4D} = \frac{1}{2W}
\begin{pmatrix}
q_2\cos q_1 t + q_1\cos q_2 t & -a(\sin q_1 t + \sin q_2 t) & a(\cos q_1 t - \cos q_2 t) & \mp q_2\sin q_1 t \pm q_1\sin q_2 t \\
a(\sin q_1 t + \sin q_2 t) & q_1\cos q_1 t + q_2\cos q_2 t & q_2\sin q_2 t - q_1\sin q_1 t & \pm a(\cos q_2 t - \cos q_1 t) \\
a(\cos q_2 t - \cos q_1 t) & q_1\sin q_1 t - q_2\sin q_2 t & q_1\cos q_1 t + q_2\cos q_2 t & \mp a(\sin q_1 t + \sin q_2 t) \\
\pm q_2\sin q_1 t \mp q_1\sin q_2 t & \pm a(\cos q_2 t - \cos q_1 t) & \pm a(\sin q_1 t + \sin q_2 t) & q_2\cos q_1 t + q_1\cos q_2 t
\end{pmatrix}
\tag{59}
$$

with the abbreviations $W^2 := (a^2 + b^2)/4$ and $q_{1;2} := W \mp b/2$.

The propagation rule for the case where $\hat{A}$ was the initial state can be read from the first column of $\mathbf{U}_{4D}$ in Eq. (59):

$$
\hat{A} \xrightarrow{\hat{H}\cdot t} \hat{A}\cdot\frac{q_2\cos q_1 t + q_1\cos q_2 t}{2W} + \hat{B}\cdot\frac{a(\sin q_1 t + \sin q_2 t)}{2W} + \hat{C}\cdot\frac{a(\cos q_2 t - \cos q_1 t)}{2W} \pm \hat{D}\cdot\frac{q_2\sin q_1 t - q_1\sin q_2 t}{2W}
\tag{60}
$$

In some cases, another form of this propagation formula may be appropriate for use, which is obtained from Eq. (60) by applying some trigonometric rules:

$$
\hat{A} \xrightarrow{\hat{H}\cdot t} \hat{A}\cdot\left(\cos(Wt)\cos\frac{bt}{2} + \frac{b}{2W}\sin(Wt)\sin\frac{bt}{2}\right) + \hat{B}\cdot\frac{a}{W}\sin(Wt)\cos\frac{bt}{2}
\tag{61}
$$

$$
-\hat{C}\cdot\frac{a}{W}\sin(Wt)\sin\frac{bt}{2} \mp \hat{D}\cdot\left(\cos(Wt)\sin\frac{bt}{2} + \frac{b}{2W}\sin(Wt)\cos\frac{bt}{2}\right)
\tag{62}
$$

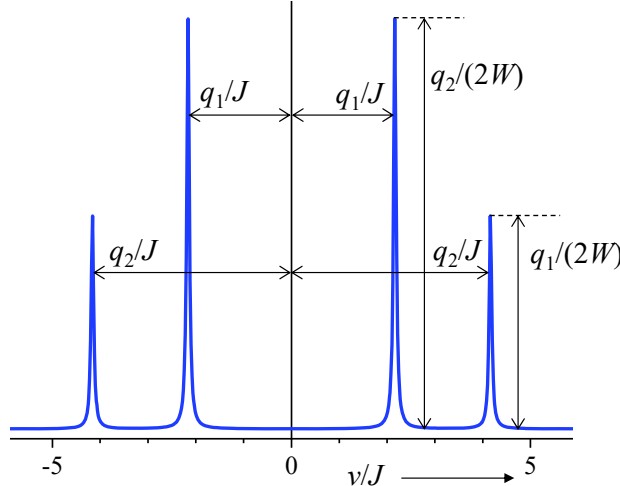

**Figure 3.** Connection between the two oscillation frequencies $q_1$ and $q_2$ in the 4D problem, example AB spin system with $\Delta\nu = 3J$.

### 4.4.2 Examples

– AB spinsystem, $J$ coupling; distance of the two lines: $\Delta\nu := \Delta\omega/(2\pi)$, the spectrometer frequency is assumed to be set
at the midpoint between the two resonances:

$$\hat{I}_{1x} + \hat{I}_{2x} \quad \xrightarrow{\;2\pi J \hat{I}_{1z} \hat{I}_{2z}\, t\;} \quad \left(\hat{I}_{1x} + \hat{I}_{2x}\right) \cdot \frac{q_2 \cos q_1 t + q_1 \cos q_2 t}{W}$$
$$+ \left(\hat{I}_{1y} - \hat{I}_{2y}\right) \cdot \frac{\Delta\omega}{2} \frac{\sin q_1 t + \sin q_2 t}{W}$$
$$+ 2\left(\hat{I}_{1z} \hat{I}_{2x} - \hat{I}_{1x} \hat{I}_{2z}\right) \cdot \frac{\Delta\omega}{2} \frac{\cos q_1 t - \cos q_2 t}{W}$$
$$- 2\left(\hat{I}_{1y} \hat{I}_{2z} + \hat{I}_{1z} \hat{I}_{2y}\right) \cdot \frac{q_1 \sin q_2 t - q_2 \sin q_1 t}{W} \tag{63}$$

where $W = 2\pi\sqrt{J^2 + \Delta\nu^2}$ and $q_{1;2} = \pi\left(\sqrt{J^2 + \Delta\nu^2} \mp J\right)$. The cosine terms contain two frequencies that give four
line positions $\pm q_1$ and $\pm q_2$ symmetrically around zero after complex Fourier transformation. The intensities are given
by the prefactors of the corresponding trigonometric functions. The lower frequency oscillation has the larger prefactor,
which reflects the roof effect. See for example Abragam (1961), chapter XI, section B. In this book, positions and
intensities are calculated from transition frequencies and probabilities for the transitions between the levels. Fig. 3 shows
the relationship between $q_1$, $q_2$ and the intensities and positions of the four lines of an AB spin system.

For the case $\Delta\nu = 0$ the Hamiltonian commutes with the initial state with the consequence that the density operator
remains constant the Fourier transform of which is just a resonance at zero frequency.

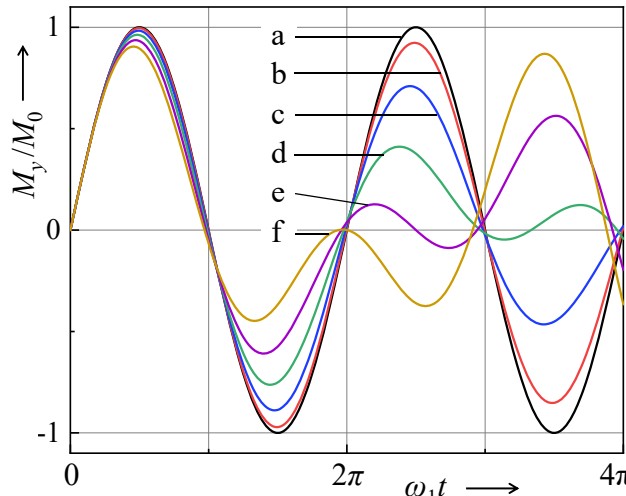

**Figure 4.** Nutation curves for different ratios of rf field strength and coupling strength. For homonuclear dipolar coupling: $D_{\mathrm{II}}/\omega_1 = $ (a): 0, (b): 0.15; (c): 0.3; (d): 0.45; (e): 0.6; (f): 0.75; for quadrupolar coupling: $\omega_Q/\omega_1 = $ (a): 0; (b): 0.2; (c): 0.4; (d) 0.6; (e): 0.8; (f): 1.

– rf irradiation onto homonuclearly coupled spins 1/2 which are initially in equilibrium (nutation):

$$
\hat{I}_{1z} + \hat{I}_{2z} \xrightarrow{\left(\hat{H}_{\mathrm{II}} + \hat{H}_{\mathrm{Ix}}\right) t} \left(\hat{I}_{1z} + \hat{I}_{2z}\right) \cdot \left(\cos Wt \cos \frac{3D_{\mathrm{II}}}{4}t + \frac{3D_{II}}{4W} \sin Wt \sin \frac{3D_{\mathrm{II}}}{4}t\right)
$$

$$
+ \left(\hat{I}_{1y} + \hat{I}_{2y}\right) \cdot \frac{\omega_{\mathrm{II}}}{W} \sin Wt \, \cos \frac{3D_{\mathrm{II}}}{4}t
$$

$$
- 2\left(\hat{I}_{1x}\hat{I}_{2z} + \hat{I}_{1z}\hat{I}_{2x}\right) \cdot \frac{\omega_{\mathrm{II}}}{W} \sin Wt \, \cos \frac{3D_{\mathrm{II}}}{4}t
$$

$$
- 2\left(\hat{I}_{1y}\hat{I}_{2x} + \hat{I}_{1x}\hat{I}_{2y}\right) \cdot \left(\cos Wt \sin \frac{3D_{\mathrm{II}}}{4}t - \frac{3D_{\mathrm{II}}}{4W} \sin Wt \cos \frac{3D_{\mathrm{II}}}{4}t\right) \tag{64}
$$

where $W = \sqrt{\omega_{\mathrm{II}}^2 + \frac{9}{16}D_{\mathrm{II}}^2}$.

This propagation formula describes the effect of a limited power rf pulse on the equilibrium magnetization. The time evolution of the prefactor of $\hat{I}_{1y} + \hat{I}_{2y}$ is shown in Fig. 4.

Comments on Eqn. (64):

– As the coupling frequency increases, so does the nutation frequency $W$. However, this oscillation is modulated by half of the dipolar frequency, see Fig. 4.

– As a consequence, the $\pi$ and $\pi/2$ conditions for achieving maximum and zero y magnetization, respectively, are modified with respect to the coupling-free case. Similar to Equation (65), this results in

$$
\tau_{\pi/2} = \frac{\pi}{2\sqrt{\omega_{\mathrm{II}}^2 + \frac{9}{16}D_{\mathrm{II}}^2}} \; ; \qquad \tau_{\pi} = \frac{\pi}{\sqrt{\omega_{\mathrm{II}}^2 + \frac{9}{16}D_{\mathrm{II}}^2}} \tag{65}
$$

– Nutation under quadrupolar interaction, spin 1:

$$\hat{I}_z \xrightarrow{\left(\hat{H}_Q+\hat{H}_{Ix}\right)t} \hat{I}_z \cdot \left(\cos\frac{Wt}{2}\cos\frac{\omega_Q t}{2} + \frac{\omega_Q t}{W}\sin\frac{Wt}{2}\sin\frac{\omega_Q t}{2}\right) + \hat{I}_y \cdot \frac{2\omega_1}{W}\sin\frac{Wt}{2}\sin\frac{\omega_Q t}{2}$$

$$- \left(\hat{I}_x\hat{I}_z + \hat{I}_z\hat{I}_x\right)\cdot\frac{2\omega_1}{W}\sin\frac{Wt}{2}\sin\frac{\omega_Q t}{2} + \left(\hat{I}_y\hat{I}_x + \hat{I}_x\hat{I}_y\right)\cdot\left(\cos\frac{Wt}{2}\sin\frac{\omega_Q t}{2} + \frac{\omega_Q t}{W}\sin\frac{Wt}{2}\cos\frac{\omega_Q t}{2}\right) \quad (66)$$

with $W = \sqrt{4\omega_1^2 + \omega_Q^2}$. This is consistent with the findings of Bloom et al. (1980), Barbara et al. (1986) and Vega and Luz (1987). Again, the result is that nutation occurs faster than $\omega_{1I}$ if there is an additional interaction.

## 4.5 Example for a 5D subspace

We consider cross polarization under finite rf power, assuming that the Hartmann-Hahn condition is satisfied: $\omega_{1I} = \omega_{1S} =: \omega_1$. Other than in the corresponding 3D examples shown above, the initial state consists of a transversally polarized $S$ spin and a depolarized $I$ spin, i.e. $\hat{\rho}(0) = \hat{S}_x$:

$$\hat{S}_x \xrightarrow{(\hat{H}_{Ix}+\hat{H}_{Sx}+\hat{H}_{IS})\cdot t} \frac{1}{2}\left[\left(\cos D_{IS}t + \frac{4\omega_1^2 + D_{IS}^2\cos qt}{q^2}\right)\cdot\hat{S}_x\right.$$

$$- \left(\frac{D_{IS}}{q}\sin qt + \sin D_{IS}t\right)\cdot 2\hat{I}_z\hat{S}_y \quad + \quad 4\frac{D_{IS}\omega_1}{q^2}(1-\cos qt)\left(\hat{I}_z\hat{S}_z - \hat{I}_y\hat{S}_y\right)$$

$$\left. + \left(\frac{D_{IS}}{q}\sin qt - \sin D_{IS}t\right)\cdot 2\hat{I}_y\hat{S}_z \quad + \quad \left(-\cos D_{IS}t + \frac{4\omega_1^2 + D_{IS}^2\cos qt}{q^2}\right)\cdot\hat{I}_x\right] \quad (67)$$

Eq. (67) describes the evolution of the magnetization of the initially polarized spin. That of the other spin evolves as given in Eq. (68):

$$\hat{I}_x \xrightarrow{(\hat{H}_{Ix}+\hat{H}_{Sx}+\hat{H}_{IS})\cdot t} \frac{1}{2}\left[\hat{S}_x\cdot\left(-\cos D_{IS}t + \frac{4\omega_1^2 + D_{IS}^2\cos qt}{q^2}\right)\right.$$

$$- \left(\frac{D_{IS}}{q}\sin qt - \sin D_{IS}t\right)\cdot 2\hat{I}_z\hat{S}_y \quad + \quad 4\frac{D_{IS}\omega_1}{q^2}(1-\cos qt)\left(\hat{I}_z\hat{S}_z - \hat{I}_y\hat{S}_y\right)$$

$$\left. - \left(\frac{D_{IS}}{q}\sin qt + \sin D_{IS}t\right)\cdot 2\hat{I}_y\hat{S}_z \quad + \quad \left(\cos D_{IS}t + \frac{4\omega_1^2 + D_{IS}^2\cos qt}{q^2}\right)\cdot\hat{I}_x\right] \quad (68)$$

## 4.6 Examples for a 6D subspace

– HH-CP experiment with finite rf power, deviation from the Hartmann-Hahn condition, the initial state consists of a transversally polarized $S$ spin and a depolarized $I$ spin, i.e. $\hat{\rho}(0) = \hat{S}_x$. Using the variables defined in subsection 4.3.5

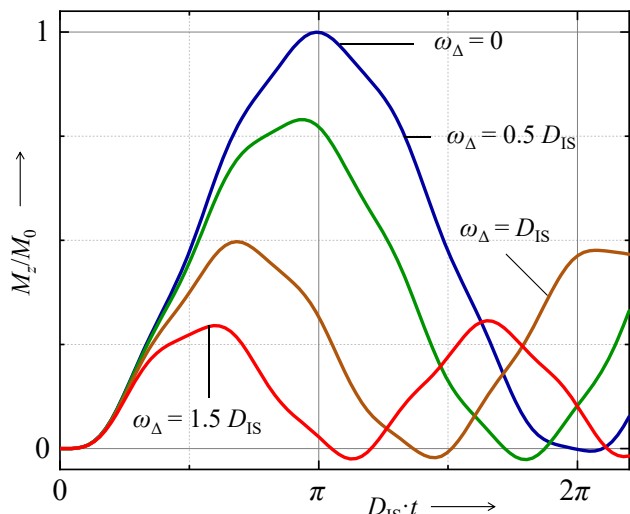

**Figure 5.** Examples for cross-polarization buildup curves for the $I$ spins in an ensemble of spin pairs $I, S$ corresponding to Eqn. (69) with $\omega_\text{ø} = 3D_\text{IS}$. The deviation $\omega_\Delta$ from the Hartmann-Hahn condition is varied.

we obtain

$$
\hat{S}_x \xrightarrow{(\hat{H}_{\text{Ix}} + \hat{H}_{\text{Sx}} + \hat{H}_{\text{IS}}) \cdot t} \hat{S}_x \cdot \frac{1}{2} \left( \frac{\omega_\Delta^2 + D_\text{IS}^2 \cos q_\Delta t}{q_\Delta^2} + \frac{4\omega_\text{ø}^2 + D_\text{IS}^2 \cos q_\text{ø} t}{q_\text{ø}^2} \right)
$$

$$
- \hat{I}_z \hat{S}_y \cdot D_\text{IS} \left( \frac{\sin q_\Delta t}{q_\Delta} + \frac{\sin q_\text{ø} t}{q_\text{ø}} \right)
$$

$$
+ 2\hat{I}_z \hat{S}_z D_\text{IS} \left[ \frac{\omega_\Delta}{2 q_\Delta} (1 - \cos q_\Delta t) + \frac{\omega_\text{ø}}{q_\text{ø}} (1 - \cos q_\text{ø} t) \right]
$$

$$
+ 2\hat{I}_y \hat{S}_y D_\text{IS} \left[ \frac{\omega_\Delta}{2 q_\Delta} (1 - \cos q_\Delta t) - \frac{\omega_\text{ø}}{q_\text{ø}} (1 - \cos q_\text{ø} t) \right]
$$

$$
+ \hat{I}_y \hat{S}_z \cdot D_\text{IS} \left( \frac{\sin q_\Delta t}{q_\Delta} - \frac{\sin q_\text{ø} t}{q_\text{ø}} \right)
$$

$$
+ \hat{I}_x \cdot \frac{1}{2} \left( \frac{4\omega_\text{ø}^2 + D_\text{IS}^2 \cos q_\text{ø} t}{q_\text{ø}^2} - \frac{\omega_\Delta^2 + D_\text{IS}^2 \cos q_\Delta t}{q_\Delta^2} \right) \tag{69}
$$

Fig. 5 shows the build-up curve for the $I$ magnetization for the case where it was initially unpolarized, and $S$ was polarized. For zero deviation from the Hartmann-Hahn condition, the curve still looks similar to the squared sine derived for infinite rf power (Eq. (53)). Small deformations result from the fact that the rf power is finite. An increasing deviation from the Hartmann-Hahn condition leads to a strong loss of the polarization-transfer efficiency, in addition to further deviation from the ideal curve.

– HHCP of an $I$ spin from two $S$ spins: The Hamiltonian is the same as in 4.3.4, and we consider the problem in the doubly-rotating frame. In this example, however, we want to follow the time evolution of all three spins individually which could not be separated in the 3D example. The propagation rules for all three spins can be found in the SI. Here is

the propagation rule for the case that the system starts with polarized S spins, i.e. $\hat{A} = \hat{S}_{1z} + \hat{S}_{2z}$ and $\hat{H} = \hat{H}_{\text{HH2}}$:

$$\hat{S}_{1z} + \hat{S}_{2z} \quad \xrightarrow{\hat{H}_{\text{HH2}} \cdot t} \quad \hat{I}_z \cdot \sin^2 \frac{qt}{2} + \left(\hat{S}_{1z} + \hat{S}_{2z}\right) \cdot \frac{1 + \cos^2 \frac{qt}{2}}{2} + \left(\hat{S}_{1z} - \hat{S}_{2z}\right) \cdot \frac{D_2^2 - D_1^2}{2q^2} \sin^2 \frac{qt}{2}$$

$$+ \left(\hat{I}_x \hat{S}_{1y} - \hat{I}_y \hat{S}_{1x}\right) \frac{D_1}{q} \sin qt + \left(\hat{I}_x \hat{S}_{2y} - \hat{I}_y \hat{S}_{2x}\right) \frac{D_2}{q} \sin qt - \hat{I}_z \left(\hat{S}_{1x} \hat{S}_{2x} - \hat{S}_{1y} \hat{S}_{2y}\right) \cdot 4 \frac{D_1 D_2}{q^2} \sin^2 \frac{qt}{2} \tag{70}$$

with $q = \sqrt{D_1^2 + D_2^2}$. This propagation rule describes the oscillatory polarization exchange between the spin polarizations and three unobservable states on the one hand, and the oscillatory polarization transfer between the spins on the other hand, as it may happen for example in $^{13}\text{CH}_2$ or in $^{15}\text{NH}_2$ groups.

## 5 Outlook: Time-dependent Hamiltonians and Liouvillians

In order to obtain the propagator matrices and the propagation formulae of the respective Liouvillians, it was assumed in the above sections that the interactions and thus also the Hamiltonian and the Liouvillian are time-invariant. However, the method can be extended to situations where the interaction constants vary with time, either fluctuating due to thermal motion or periodically due to sample spinning. The commutator equations are also valid, i.e. steps 1 and 2 of the given procedure can be performed to obtain the relevant subspace and the Liouvillian matrix associated with that subspace. In other words, the Liouvillian matrices obtained for the above examples (see SI) can also be used in the time-dependent situations. The Liouville-von Neumann equation now belongs to a system of linear differential equations, but with time-dependent coefficients. There is no general scheme for their integration. The propagator matrix is no longer the matrix exponential of $-i\mathbf{L}t$.

However, in some cases solutions are possible in the following ways: (i) Use of the time-averaged Liouvillian for an exact solution when $\mathbf{L}$ depends linearly on a single time-dependent parameter, (ii) Use of the Shirley method, based on the Floquet theorem, also for an analytical solution.

The first way can be justified as follows: For the Liouville-vonNeumann equation in matrixform, one can try to find an effective Liouvillian matrix $\mathbf{L}_{\text{eff}}(t)$, which is defined as that matrix which is constant in the interval $(0,t)$ and which has the same effect as the actual $\mathbf{L}$, i.e. $\mathbf{U}(t) = \exp(-i\mathbf{L}_{\text{eff}}t)$ in matrix notation. This can be done by the Magnus expansion (Magnus, 1954):

$$\mathbf{L}_{\text{eff}} = \int_0^t \mathbf{L}(t_1)\, dt_1 + \frac{1}{2} \int_0^t [\mathbf{L}(t_1), \mathbf{L}(t_2)]\, dt_1 dt_2 + O(\mathbf{L}^3) \tag{71}$$

Although the convergence radius of this series is rather small (Maricq, 1982), it can nevertheless be used to check whether the effective Liouvillian can be replaced by the time-averaged Liouvillian (first term on the right-hand side of Eq. (71)). If the Liouvillian commutes at all times with itself, all higher-order terms vanish, only the zeroth order term survives. In this case, the effective Liouvillian is equal to the time-averaged Liouvillian. This happens besides the case of a constant Liouvillian (see above), if the Liouvillian matrix can be written as product of a scalar function $\lambda(t)$ with a constant matrix:

$$\mathbf{L}(t) = \lambda(t)\,\mathbf{A} \qquad \rightarrow \qquad \mathbf{L}_{\text{eff}} = \mathbf{A} \int_0^t \lambda(t_1)\, dt_1 \tag{72}$$

In this case, the propagator matrix is exactly the matrix exponential

$$
\mathbf{U}(t) = \exp\left[ -i\mathbf{A} \int_0^t \lambda(t_1)\, dt_1 \right]
\tag{73}
$$

This concerns all 2D cases. Eq. (31) becomes

$$
\mathbf{L_{2D}} = \lambda(t) \begin{pmatrix} 0 & -i \\ i & 0 \end{pmatrix} \;;\qquad
\mathbf{U_{2D}} = \exp\left( -i \int_0^t \mathbf{L_{2D}}(t_1)\, dt_1 \right) = \begin{pmatrix} \cos \int_0^t \lambda(t_1)\, dt_1 & -\sin \int_0^t \lambda(t_1)\, dt_1 \\ \sin \int_0^t \lambda(t_1)\, dt_1 & \cos \int_0^t \lambda(t_1)\, dt_1 \end{pmatrix}
\tag{74}
$$

The transformation to propagation formulae gives:

$$
\hat{A} \xrightarrow{\;\hat{H}\,t\;} \hat{A}\cos \int_0^t \lambda(t_1)\, dt_1 + \hat{B}\sin \int_0^t \lambda(t_1)\, dt_1 \quad;
$$

$$
\hat{B} \xrightarrow{\;\hat{H}\,t\;} \hat{B}\cos \int_0^t \lambda(t_1)\, dt_1 - \hat{A}\sin \int_0^t \lambda(t_1)\, dt_1
\tag{75}
$$

Eqs. (75) can be regarded as a rigorous extension of the POF to time-dependent systems, such as those caused by thermal motion or by sample spinning. For higher dimensional cases, equation (72) can only be satisfied if the two parameters $a$ and $b$ are equal. While for the 4D, 5D and 6D cases this would mean very special situations, the 3D case includes with $a = b$ the important case of cross polarization. Then Eq. (38) changes to

$$
\mathbf{L_{3D}} = a(t) \begin{pmatrix} 0 & -i & 0 \\ i & 0 & -i \\ 0 & i & 0 \end{pmatrix}
\tag{76}
$$

if $b$ is replaced by $a$. In the subsequent propagation formulae we have to substitute $qt$ by $\int_0^t q(t_1)\, dt_1$ with $q(t) = a(t)\sqrt{2}$. This means that we obtain an exact solution if we replace the arguments of the trigonometric functions by integrals in the following examples from above:

- Eqs. (32), (35) and (52): $D_{\mathrm{II}}t$ by $\int_0^t D_{\mathrm{II}}(t_1)\, dt_1$

- Eqs. (33), (36) and (53): $D_{\mathrm{IS}}t$ by $\int_0^t D_{\mathrm{IS}}(t_1)\, dt_1$

- Eq. (34): $\omega_{\mathrm{Q}}t$ by $\int_0^t \omega_{\mathrm{Q}}(t_1)\, dt_1$

However, there is no general recipe for all other cases. In some cases, the Shirley method using the Floquet theorem, which is applied e.g. for numerical calculations of spin systems under MAS, may also be successful for obtaining analytical expressions. This will be the subject of a forthcoming paper.

# 6 Conclusions

Repeated application of the commutator of the Hamiltonian with the initial density operator gives a system of operator equations, the coefficient matrix of which can be used to establish a propagation rule for the spin system. This has been demonstrated in this paper with some examples. A more detailed analysis shows that the commutator relations define subspaces which are both Liouvillian invariant and superpropagator invariant. Therefore, the density operator propagates in such a subspace without leaving it. If its dimension is small enough, analytical expressions for the propagation law can be obtained.

The relevant subspace for a given problem is determined by the Hamiltonian and by the initial state. If the operator characterizing the initial state changes, then the new subspace is the same as the previous one if and only if the new initial-state operator is an element of the previous subspace, otherwise the two subspaces have no intersection.

The set of problems can be divided into classes with respect to the dimension of the subspaces. Problems of the 2D class can be easily treated by propagation formulae similar to those of the well-known product-operator formalism. The propagation

formulae for the 3D and 4D classes are given in this paper, and an example is given for the 5D and 6D cases, respectively. If necessary, the method introduced and explained here can also be applied to cases with higher dimensions. The application examples demonstrate the same mathematical structure of some physically different problems.

In addition, this treatment can be applied to pulse sequences in a manner similar to the POF. In some cases, an algebraic language program may be helpful. Even numerical computations can use this framework by starting the numerical computation

on the basis of existing analytical relations. In the $N$-dimensional wavefunction space, the Liouville-vonNeumann equation corresponds to a system of $N^2$ differential equations, which is significantly larger than that obtained by the dimension reduction with the method presented here. It may be advantageous to first apply this method on an analytical basis before starting the numerical implementation.

If the strength of any of the considered interactions depends on time, the first two steps of this method can be applied like-

530 wise. Step 3 (matrix exponentialization), however, has to be modified because there is no general recipe for solving a system of differential equations with time-dependent coefficients. In some cases, the time-averaged Liouvillian is suitable to get an exact solution of the problem. Even for numerical calculations it can be helpful to start with systems of differential equations containing a reduced number of equations.

## Appendix A: Norms of spin operators

Consider a system of $N$ spins 1/2. As proved in the SI (section 1.3), the norm of a product of the Cartesian components of $n$ spins from this $N$-spin-system (one operator for each spin) amounts to $2^{(N/2)-n}$. Furthermore, for the sum of two orthogonal operators $\hat{A}$ and $\hat{B}$ holds $\|\hat{A}+\hat{B}\| = \sqrt{\|\hat{A}\|^2 + \|\hat{B}\|^2}$. From this, we can derive the following rules ($\alpha, \beta, \gamma, \delta \in \{x, y, z\}$):

- For single spins $I = \frac{1}{2}$: $\|\hat{I}_\alpha\| = \frac{1}{\sqrt{2}}$;

- For spin pairs $I = \frac{1}{2}$, $S = \frac{1}{2}$: $\|\hat{I}_\alpha\| = \|\hat{S}_\alpha\| = 1$ ; $\quad \|\hat{I}_\alpha \hat{S}_\beta\| = \frac{1}{2}$ ; $\quad \|\hat{I}_\alpha \hat{S}_\beta \pm \hat{I}_\gamma \hat{S}_\delta\| = \frac{1}{\sqrt{2}}$

– For spin triples $I = \frac{1}{2}$, $J = \frac{1}{2}$, $S = \frac{1}{2}$:

$$\|\hat{I}_\alpha\| = \|\hat{J}_\alpha\| = \|\hat{S}_\alpha\| = \sqrt{2}\,; \quad \|\hat{I}_\alpha \hat{S}_\beta\| = \frac{1}{\sqrt{2}}\,; \quad \|\hat{I}_\alpha \hat{S}_\beta \pm \hat{I}_\gamma \hat{S}_\delta\| = 1\,; \quad \|\hat{I}_\alpha \hat{J}_\beta \hat{S}_\gamma\| = \frac{1}{2\sqrt{2}}$$

Similarly, for a single spin $I = 1$ can be obtained:

$$\|\hat{I}_\alpha\| = \sqrt{2}\,; \quad \|\hat{I}_\alpha \hat{I}_\beta + \hat{I}_\beta \hat{I}_\alpha\| = \sqrt{2}$$

## Appendix B: Hamiltonians used in the main part and in the SI

– Interaction of the I and S spins with the rf field, strengths $\omega_{1I}$ ans $\omega_{1S}$, respectively, assumed to be constant and parallel to the x axis in the rotating frame:

$$\hat{H}_{Ix} = -\omega_{1I}\hat{I}_x\,; \qquad \hat{H}_{Sx} = -\omega_{1S}\hat{S}_x \tag{B1}$$

Unless otherwise indicated, the irradiation occurs at the Larmor frequency of the respective spins. The rf phase is always such that $\mathbf{B}_1$ is parallel to the $x$ axis of the rotating frame, but the calculations can easily be modified to include other
directions of the rf field.

   – Homonuclear dipolar interaction between spin $\mathbf{I}_1$ and spin $\mathbf{I}_2$ (secular part):

$$\hat{H}_{II} = D_{II}\left(\hat{\mathbf{I}}_1\hat{\mathbf{I}}_2 - 3\hat{I}_{1z}\hat{I}_{2z}\right) \tag{B2}$$

where $D_{II} = (\mu_0\gamma_I^2\hbar/(4\pi r_{II}^3))\left(3\cos^2\theta_{II} - 1\right)$, $r_{II}$ is the length of the vector connecting both spins and $\theta_{II}$ is the angle of this vector with the external magnetic field, $\gamma_I$ is the gyromagnetic ratio of the $I$ spins and $\mu_0$ is the permeability of the
vacuum.

   – Heteronuclear dipolar interaction between spin $\mathbf{I}$ and spin $\mathbf{S}$ (secular part):

$$\hat{H}_{IS} = -D_{IS} \cdot 2\hat{I}_z\hat{S}_z \tag{B3}$$

where $D_{IS} = \frac{\mu_0}{4\pi}\frac{\hbar\gamma_I\gamma_S}{r_{IS}^3}\left(3\cos^2\theta_{IS} - 1\right)$, $r_{IS}$ is the length of the vector connecting both spins and $\theta_{IS}$ is the angle of this vector with the external magnetic field, $\gamma_S$ is the gyromagnetic ratio of the $S$ spins. The case of $J$ coupling between
unlike spins is mathematically equivalent to this; the corresponding formulae can be obtained by replacing $D_{IS}$ by $-\pi J$.

   – Matched Hartmann-Hahn cross polarization between spin $I$ and spin $S$ in the doubly-rotating frame (Hartmann and Hahn, 1962):

$$\hat{H}_{HH} = -D_{IS} \cdot \left(\hat{I}_x\hat{S}_x + \hat{I}_y\hat{S}_y\right) \tag{B4}$$

assuming $D_{IS} \ll \omega_{1I}, \omega_{1S}$ so that rapidly oscillating terms can be neglected.

– Interaction of first order of the nuclear quadrupole moment with the electric field gradient (secular part):

$$\hat{H}_Q = \omega_Q \cdot \left( \hat{I}_z^2 - \frac{I(I+1)}{3} \right) \tag{B5}$$

where $\omega_Q$ is the quadrupolar frequency.

– Resonance offset $\Delta\omega$ in the rotating frame:

$$\hat{H}_\Delta = -\Delta\omega \, \hat{I}_z \tag{B6}$$

The coupling frequencies are assumed to be unique, i.e. no powder averaging unless otherwise stated. It is also assumed that the coupling frequencies are constant in time, i.e. thermal motion or sample rotation is not considered.

*Author contributions.* The author G.H. developed the method, used it to derive the examples given here and compared them with known equations, if any.

*Competing interests.* There are no competing interests related to this article.

*Acknowledgements.* The author would like to thank Kay Saalwächter for advice on the preparation of this manuscript.

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
