# Peer review of "Analytical expressions for time evolution of spin systems affected by two or more interactions"

_Magnetic Resonance, 2024_

## Referee Comment (RC1)

**Reviewer Report:** Hempel, Analytical expressions for time evolution of spin systems affected by two or more interactions

Hempel introduces a method by which one can reduce the state space of exponentially-scaling spin systems and thereby derive analytical expressions for the time evolution. The method hinges on well known properties of Lie algebras, particularly that products of any two elements of a Lie group generate another member of that Lie group. This allows the author to generate the subspace of operators required to fully describe time evolution in specific cases given an initial starting state, thus permitting dimensionality reduction of the system. The author demonstrates for a variety of different cases how a mD space may be reduced to an nD space and derives analytical means for time evolution in those systems.

I would like to start by stating that the paper is generally laid out well and provides a thorough description of the development along with a diverse set of examples that support the theoretical tool being developed, and in general, I believe this is of worth to the Magnetic Resonance community.

My major criticism of the paper is that, in its current form, I cannot think of the intended target audience for this paper. If the work is intended as an extension to the typical product operator formalism, which is popular among those who are new to the field, I believe it lacks sufficient background information and explanation of the various cases that are examined to be helpful to that community - I believe it would be more worthwhile to examine fewer cases in the main text for each dimensionality but discuss them more thoroughly. At this level, analytical expressions can be useful to talk about pulse sequence elements, but they quickly lose their applicability as soon as the experimental complexity becomes more than a few pulses. Furthermore, the use of analytical expressions is difficult to motivate in the time when running these simulations (which in this paper correspond to, at largest, 4-dimensional matrices in the Hilbert space) is trivial. Hence, I believe that it could be expanded into an excellent continuation of what is introduced in the conventional product operator formalism.

I also think that the method is interesting to those who do numerical simulations of spin systems, as this lays out a dimensionality reduction technique that requires no approximation of the system. I acknowledge that this is no longer a topic in the direction of analytical expressions, but I believe it is a paper that is well laid-out for a theoretician in magnetic resonance that would guide them through a dimensionality reduction approach. A section that discusses this possibility might be of interest, although I don't think it is required for publication.

Broadly, I would like to also ask the author to comment in the article about when this technique can be applied in general, as practical applications are limited to A) Hamiltonians that are time-independent or B) Hamiltonians that can be cast in a frame where they become time-independent, either by use of Average Hamiltonian Theory or other analytical methods like toggling/rotating/interaction frames.

During the process of writing this report, I do have a following major concern with the formulation as it is written. In equation 19, the author presents the formulation of the product $\mathbf{L} \cdot \boldsymbol{\rho}$ as the left-handed multiplication of the Liouvillian with the basis vector of operators. This led to my comment that equation 22 has index typos in it, as the dot product $\mathbf{U} \cdot \boldsymbol{\rho}$ (also left handed by integration of eq. 19) would generate terms such as $\hat{A}_2 U_{12}$ in the evolution of $\hat{A}_1$. In equation 35, the author says that the propagation rules are obtained from the *columns* of the propagator, however that is not the case if the author has formulated the theory with left-handed multiplication. The propagation rules are obtained from the *rows* of the propagator. At the end of the day, this is only rectified by the fact that the time-symmetry of quantum mechanics allows for this (up to a definition of a phase). As such, either I have

misinterpreted how the evolution is calculated ($\mathbf{U} \cdot \boldsymbol{\rho}$), which is unclear given the ambiguous notation in the appropriate sections, or the author has accidentally taken the wrong set of elements from the propagator.

Specifically, I note the following points:

- *Page 2 line 31:* "dipol-dipol" is a typo.

- *Page 4 line 79:* Minor, but the author should use proper typesetting for dot products ($\mathbf{A}^{\dagger} \cdot \mathbf{B}$ instead of $\mathbf{A}^{\dagger} . \mathbf{B}$). This is a problem throughout page 4, but should be consistent in the entire article.

- *Page 5 line 122:* "on each operator A" missing the hat on the $\hat{A}$.

- *Page 6 line 152:* I find the notation that is introduced overly confusing for this section, particularly, terms such as the $\lambda_{11}\hat{A}_1$ term in equation 16 are by definition zero, a point that is made in the very line in question. Thus it leaves the reader somewhat confused as to why equation 16 would contain this term. The same can be said of the $\lambda_{22}\hat{A}_2$ term in equation 17. The author should either specifically state that the coefficients $\lambda_{nn} = 0$ by definition or drop them from the equations.

- *Page 7 line 77:* "The action of the Liouvillian on any $\hat{A}_i$ ($i \in 1..N$ leads to a linear combination of the $\hat{A}_i$". Firstly, I believe there is a parentheses that is missing in after the $N$. However, this is a general result of the operators for spin systems being part of a Lie algebra and is well known. For a detailed explanation, I recommend *Spin* by Ilya Kuprov.

- *Page 8 line 196:* The author has typos in the indices in equation 22. For instance, one should find the term $\hat{A}_2 \cdot U_{12}$ in the evolution of $\hat{A}_1$ (line 196), if I have interpreted correctly that the dot product being calculated is $\mathbf{U} \cdot \begin{bmatrix} \hat{A}_1 & \hat{A}_2 & \cdots & \hat{A}_N \end{bmatrix}^{\mathrm{T}}$, which is also unclear as it is not stated and only indicated through the ambiguous arrow with a $\hat{H} \cdot t$ decorated over it. Furthermore, please only use dot products when they are actually between two objects of rank-1 or higher, otherwise it is ambiguous what is intended.

- *Page 8 line 198:* "recompose the matrix exponential for each new situation". It is unclear what the author means by "each new situation". Please elaborate or be specific.

- *Page 9 line 227:* The author introduces in the case of a 2D subspace the operator basis

$$\{\hat{A}_i\} = \left\{ \left( \hat{I}_{1x} + \hat{I}_{2x} \right), \ 2 \left( \hat{I}_{1z}\hat{I}_{2y} + \hat{I}_{1y}\hat{I}_{2z} \right) \right\}$$

which is a linear combination of the operators that one would typically encounter when exploring a 2-spin system for the first time (formed by the permutations of $\{\hat{E}, \hat{I}_x, \hat{I}_y, \hat{I}_z\}$). As such, those unfamiliar with why those operators may be linearly combined would likely be confused by this, as it is a non-intuitive basis to build. The author should discuss how, when this procedure is carried out in the native operator basis where the elements are instead

$$\{\hat{A}_i\} = \left\{\hat{I}_{1x},\ \hat{I}_{2x},\ 2\hat{I}_{1z}\hat{I}_{2y},\ 2\hat{I}_{1y}\hat{I}_{2z}\right\}$$

how the appropriate matrices appear and how one can further reduce this 4D case to a 2D case. This is a critical part of the procedure that is missing from the article, and is not unique to this instance.

- *Page 10 line 235:* If the author wishes to discuss the case of cross polarization, it should be noted that the author has rearranged the initial spin state into $\hat{S}_z - \hat{I}_z$ and $\hat{S}_z + \hat{I}_z$, the latter of which does not evolve and the former which dictates the polarization transfer. This goes along the lines of explaining the operator basis that is used, as without this, the problem would be at least 3D (if one already has collected the zero-quantum terms into one basis state).

- *Page 10 lines 241:* "The procedure described above reaches the cancellation condition after three commutators of the kind". This language is ambiguous and makes it sound like equation 33 is a generic result to the method and not specific for the types of systems that belong to the 3D case.

- *Page 10 line 246:* It would be helpful if the author showed how this was calculated.

- *Page 12 line 292:* "In resonance" should be "on resonance".

- *Page 13 line 296:* "In resonance" should be "on resonance".

- *Page 14 line 321:* It would be helpful to explain what the LG condition is to the reader.

- *Page 15 line 341:* "the more the smaller the rf power" clumsy phrasing.

---

## Referee Comment (RC2)

**General comments**

The author provides an exhaustive collection of analytical expressions for cases, in which the spin density operator evolves in a low-dimensional subspace of the Liouville space. Although the theory behind the procedure is not new and some of the examples are known from literature, the "completeness" of the work presented in the manuscript makes it a very useful source of information for researchers who want to go beyond numerical "black box" calculations and want to get a better understanding of the evolution of spin systems. I think the manuscript is very interesting both for experts and for newcomers in the field of magnetic resonance and I would like to see this contribution published.

**Specific comments**

To make the manuscript easier to understand for newcomers I suggest to include a bit more background information. Some essential background knowledge is taken for granted, making the understandability sometimes a bit difficult. In my opinion, it should be explained explicitly what is behind the "arrow notation" of the propagation rules introduced in equation (1) and (2), for example, by providing an equation like

$$\hat{\rho}(t) = \exp(-i\hat{H}t)\hat{\rho}(0)\exp(i\hat{H}t).$$

By expanding the exponentials in this equation the occurence of the multiple commutators can be readily explained. Without this, the mentioning of repeated calculations of commutators comes "out of the blue", at least for less experienced readers.

I also have a problem understanding section 1.1 of the SI: I find it difficult to bring the equation and the text above it together — perhaps the text can be rephrased more clearly. I guess the meaning of the arrow $\mapsto$ is *maps to*. It may also help to explicitly explain the different symbols used for abstract operators (such as $\hat{\rho}$) and their matrix representations (such as $\boldsymbol{\rho}$).

**Technical issues**

Overall, the text is diligently written but it is no surprise that such a comprehensive document comes with some errors. In the following, both the errors I found (not having read every sentence or equation!), some suggestions and minor questions are listed.

1. page 1, line 22ff: The introductory example mentioned here is propagation of transversal magnetization of spin $I = 1/2$ (which is a good choice) but the following equation (1) and the text on page 2, line 25, contain $I_z$ instead of $I_x$. In eq. (1), both occurrences of $I_z$ should be replaced.

2. page 2, line 28: I suggest to replace *in this case* by *in this example*.

3. page 2, line 31: Typo in *dipol-dipol*

4. same line: I suggest to add *or* before *cross polarization.*

5. page 3, lines 50–51: I suggest to replace *independent of the dimension of the latter* by *although the Liouville space has a much larger dimension.*

6. page 3, line 52: I wonder if the statement *"However, condition (3) cannot be fulfilled if more than one interaction has to be considered"* is always true.

7. page 3, line 54: *...an initial state* $\rho_0 = \hat{I}_z$. (Shouldn't the density operator carry a ˆ?). This is one of the few instances, where the equal sign (=) is used for assigning the initial state. In most of the manuscript (including SI), assignments of special values are indicated by arrows ($\rightarrow$). I prefer the equal sign because the arrow can be misinterpreted as indication of a limiting value or, in the context of this manuscript, a time evolution.

8. page 3, line 63: I suggest to rephrase the sentence (for better understandability) and write: *...note the* $2 \times 4$ *matrix in Eq. (5) is the exponential of the* $2 \times 2$ *matrix in Eq. (6) multiplied by* $-it$ ....

9. page 3, line 67: change *formed* to *formulated*

10. page 3, line 70: change *was possible* to *is possible*

11. page 3, line 74: change *for the further work here* to *this work*

12. page 4, line 85: I think *estimating* should be changed to *calculating.* (There are more instances, where *estimate* is used instead of *calculate.* Please check.)

13. page 4, lines 86–87. I suggest to rephrase the sentence: *...but it depends on the relevant space, which is different for different numbers of spins.*

14. page 5, line 118: change estimation to *calculation*

15. page 5, line 122: The operator $\hat{A}$ is missing its hat.

16. page 6, lines 152–153: *see Example 1D-1 in the SI.* In the SI, there is no such example. A 1D subspace is mentioned in section 4.1.

17. page 7, line 172: extra *all*

18. page 7, line 177: parenthesis not closed

19. page 7, line 181: extra *above*

20. page 7, lines 181–182: Shouldn't all $N$ be replaced by $n$?

21. page 8, Eq. 22: Isn't the matrix U multiplied from the left, resulting in $\hat{A}_1 \cdot U_{11} + \hat{A}_2 \cdot U_{21} + \ldots$ (inverted indices of $U_{kl}$)?

22. page 9, line 219: I suggest *... appearing in Eq. (5) and (6).*

23. page 10, line 243: replace *estimate* by *calculate*

24. page 12, line 286: typo, it should probably read: *... with the amplitude* $\frac{D_{IS}^2}{\omega_{IS}^2 + D_{IS}^2}$.

25. page 13, line 319: replace DRKS by *doubly rotating frame* (I think it should be "doubly rotating" instead of "double rotating" everywhere.)

26. page 17, line 367: I think *... larger prefactor, which reflects the roof effect* is correct.

27. page 18, line 376: *a limited power*

28. page 20, line 406: *an I spin*

29. page 21, lines 432–433: $N$ and $n$ not clear. I think $N$ is the total number of spins, and $n$ the number of factors in the product. For clarity one should write $2^{(N/2)-n}$—if I understood it correctly.

30. page 22, line 461: What is $\omega_{1I;S}$?

31. page 23, lines 469–470: Perhaps better *... developed the method, used it to derive the examples given here and ...*?

32. page 23, line 472: no plural for *advice*

33. SI, page 4, line after (S2): instead of *Similarly* the use of *Similar to the dipolar Hamiltonian* might be more informative.

34. SI, page 4, line 4 from bottom: How about *... is parallel magnetization of spins $I_1$ and $I_2$, aligned transversal to* $\mathbf{B_0}$?

35. SI, page 5, line after 3.2.2.3: The extra punctuation mark after *Hamiltonian:* should be deleted.

36. SI, page 6, sentence before 3.2.3.2: *. . . not $-(3/2)D_{II}$ . . . (example 2D-1)* (minus-sign for completeness, wrong example number)

37. SI, page 7, line 7 (including eq.): *. . . can be detected . . .*

38. SI, page 7, change of sentence: *The cases where the relevant magnetic field strengths are not large with respect to the coupling frequency and where deviations from Hartmann-Hahn condition occur are problems . . .*

39. SI, page 7, eq. (S15) and (S16): What is $q$?

40. SI, page 8, line 1 after 3.3.2.1 Here and elsewhere: replace all *Equ.* by *Eq.*

41. SI, page 8, line 2 after (S17): replace *what* by *which*

42. SI, page 9, (S21): typo, change to *crossing*

43. SI, page 10, line 7 after (S23): Do you mean *approaches* instead of *approximates*?

44. SI, page 10, line 7 after (S23): Avoid starting the sentence with I.e., one could write *In other words, it describes . . .*

45. SI, page 10, line 3 before 3.3.3.2: Missing word: *The constant component is subject . . .*

46. SI, page 12, line 1: *approach* instead of *approximate*

Sorry, I had to stop here because I ran out of time.

---

## Author Comment (AC4)

**Reply to Reviewer Comment 2**

RC2: To make the manuscript easier to understand for newcomers I suggest to include a bit more background information. Some essential background knowledge is taken for granted, making the understandability sometimes a bit difficult. In my opinion, it should be explained explicitly what is behind the "arrow notation" of the propagation rules introduced in equation (1) and (2), for example, by providing an equation like

$$\hat{\rho}(t) = \exp\left(-i\hat{H}t\right)\hat{\rho}(0)\exp\left(i\hat{H}t\right)$$

By expanding the exponentials in this equation the occurence of the multiple commutators can be readily explained. Without this, the mentioning of repeated calculations of commutators comes "out of the blue", at least for less experienced readers.

GH: I will mention in the introduction that the arrow notation is a widely used notation at least if the product-operator formalism is applied.
Moreover, as a consequence to the discussions concerning the indices in Eq. (22) and the related SI section 1.1, I will insert into 2.1 a paragraph dealing with the connection between the propagation rules (those with an arrow) and the superoperator-density operator equations like $\hat{\rho}(t) = \hat{\hat{U}}\,\hat{\rho}_0$. This also serves to emphasise that the two forms of the representation of the time evolution take place in different spaces: The arrow notation uses an operator base, while the matrix notation uses a base of column matrices. This means that the corresponding matrices are transposed to each other because they originate from dual spaces. I hope that this will clear up the misunderstandings regarding the indexing in equation (22).
My access to the commutator equation systems comes from the requirement of having a Liouville-invariant subspace, i.e. a multiple application of the Liouvillian and therefore the commutator serves as a tool to ensure the L-invariance of the subspace. Then this criterion is used also to find such a subspace. I will emphasize this more in the subsections 3.1 ans 3.2.

RC2: I also have a problem understanding section 1.1 of the SI: I find it difficult to bring the equation and the text above it together — perhaps the text can be rephrased more clearly. I guess the meaning of the arrow 7→ is *maps to*.

GH: I agree that this section of the SI was inadequate to explain some relations from the Main Part. The purpose of this section is to prove that (i) the

Liouvillian matrix is the transposed coefficient matrix of the system of commutator equations, and (ii) the coefficient matrix of the propagation formulae is the transposed propagator matrix. I have rewritten this section completely in the form containing proposition and proof. I hope that this will enable the reader to see the purpose of the section and, most importantly, to see why some relations given in the Main Part exist. For the latter, I will insert some remarks at the relevant places in the Main Part.

RC2: It may also help to explicitly explain the different symbols used for abstract operators (such as $\hat{\rho}$) and their matrix representations (such as $\rho$).

GH: I will insert a paragraph explaining these different symbols.

**Technical issues**

1. page 1, line 22ff: The introductory example mentioned here is propagation of transversal magnetization of spin $I = 1/2$ (which is a good choice) but the following equation (1) and the text on page 2, line 25, contain $I_z$ instead of $I_x$. In eq. (1), both occurrences of $I_z$ should be replaced.

    GH: Thank you, this has been corrected.

2. page 2, line 28: I suggest to replace *in this case* by *in this example*.

    GH: This suggestion has been added.

3. page 2, line 31: Typo in *dipol-dipol*

    GH: Corrected.

4. same line: I suggest to add *or* before *cross polarization*.

    GH: This suggestion has been added.

5. page 3, lines 50–51: I suggest to replace *independent of the dimension of the latter* by *although the Liouville space has a much larger dimension*.

    GH: This suggestion has been added.

6. page 3, line 52: I wonder if the statement *"However, condition (3) cannot be fulfilled if more than one interaction has to be considered"* is always true.

    GH: I know of no counterexample, but I know of no proof. So I replace the "cannot be" with "is often not".

7. page 3, line 54: *...an initial state* $\rho_0 = \hat{I}_z$. (Shouldn't the density operator carry a ^?). This is one of the few instances, where the equal sign (=) is used for assigning the initial state. In most of the manuscript (including

SI), assignments of special values are indicated by arrows ($\rightarrow$). I prefer the equal sign because the arrow can be misinterpreted as indication of a limiting value or, in the context of this manuscript, a time evolution.

GH: I agree with that comment. At first I wanted to characterize a substitution with this arrow, but now I see the danger of confusion and will replace all arrows with equal signs. The missing hat has been added.

8. page 3, line 63: I suggest to rephrase the sentence (for better understandability) and write: *...note the* $2 \times 4$ *matrix in Eq. (5) is the exponential of the* $2 \times 2$ *matrix in Eq. (6) multiplied by* $-it$ ....

GH: I have changed my inadequate formulation to this suggestion.

9. page 3, line 67: change *formed* to *formulated*

GH: Changed.

10. page 3, line 70: change *was possible* to *is possible*

GH: Changed.

11. page 3, line 74: change *for the further work here* to *this work*

GH: Changed.

12. page 4, line 85: I think *estimating* should be changed to *calculating*. (There are more instances, where *estimate* is used instead of *calculate*. Please check.)

GH: Corrected.

13. page 4, lines 86–87. I suggest to rephrase the sentence: *...but it depends on the relevant space, which is different for different numbers of spins.*

GH: This suggestion has been added.

14. page 5, line 118: change estimation to *calculation*

GH: Corrected.

15. page 5, line 122: The operator $\hat{A}$ is missing its hat.

GH: Corrected.

16. page 6, lines 152–153: *see Example 1D-1 in the SI.* In the SI, there is no such example. A 1D subspace is mentioned in section 4.1.

GH: Yes, that needs to be corrected.

17. page 7, line 172: extra *all*

    GH: Corrected.

18. page 7, line 177: parenthesis not closed

    GH: Corrected.

19. page 7, line 181: extra *above*

    GH: Corrected.

20. page 7, lines 181–182: Shouldn't all *N* be replaced by *n*?

    GH: Yes, that needs to be corrected.

21. page 8, Eq. 22: Isn't the matrix U multiplied from the left, resulting in $\hat{A}_1 \cdot U_{11} + \hat{A}_2 \cdot U_{21} + ...$ (inverted indices of $U_{kl}$)?

    GH: The coefficient matrix of the propagation formulae is the transposed propagator matrix, so the indices in Eq. (22) are correct. This relationship was not adequately mentioned in the original manuscript incl. SI. See the reply to the reviewer's comment on SI section 1.1 above. This SI section has been completely rewritten to emphasize this relationship.

22. page 9, line 219: I suggest *...appearing in Eq. (5) and (6).*

    GH: This suggestion has been added.

23. page 10, line 243: replace *estimate* by *calculate*

    GH: This sentence has been changed due to insertion of a note that the Liouvillian is the transposed coefficient matrix of the commutator equations.

24. page 12, line 286: typo, it should probably read: *...with the amplitude* $\frac{\tilde{D}_{IS}^2}{\omega_{IS}^2 + D_{IS}^2}$ .

    GH: Corrected.

25. page 13, line 319: replace DRKS by *doubly rotating frame* (I think it should be "doubly rotating" instead of "double rotating" everywhere.)

    GH: Corrected everywhere.

26. page 17, line 367: I think *...larger prefactor, which reflects the roof effect* is correct.

    GH: Corrected.

27. page 18, line 376: *a limited power*

    GH: Corrected.

28. page 20, line 406: *an I spin*

    GH: Corrected.

29. page 21, lines 432–433: $N$ and $n$ not clear. I think $N$ is the total number of spins, and $n$ the number of factors in the product. For clarity one should write $2^{(N/2)-n}$—if I understood it correctly.

    GH: $N$ and $n$ have the meaning that the reviewer assumed. I change the sentence to include an explanation for these variables. The N/2 in the exponent is set into parentheses.

30. page 22, line 461: What is $\omega_{1I;S}$?

    GH: I have replaced this variable with mixed index by two separate terms $\omega_{1I}$ and $\omega_{1S}$.

31. page 23, lines 469–470: Perhaps better *...developed the method, used it to derive the examples given here and ...*?

    GH: This proposal was adopted.

32. page 23, line 472: no plural for *advice*

    GH: Corrected.

33. SI, page 4, line after (S2): instead of *Similarly* the use of *Similar to the dipolar Hamiltonian* might be more informative.

    GH: This proposal was adopted.

34. SI, page 4, line 4 from bottom: How about *...is parallel magnetization of spins $I_1$ and $I_2$, aligned transversal to* $\mathbf{B_0}$?

    GH: This proposal was adopted.

35. SI, page 5, line after 3.2.2.3: The extra punctuation mark after *Hamiltonian:* should be deleted.

GH: Corrected.

36. SI, page 6, sentence before 3.2.3.2: *…not −(3/2)$D_{II}$…(example 2D-1)* (minus-sign for completeness, wrong example number)

   GH: The minus is added, the example number is changed to 2D-1.

37. SI, page 7, line 7 (including eq.): *…can be detected …*

   GH: Corrected.

38. SI, page 7, change of sentence: *The cases where the relevant magnetic field strengths are not large with respect to the coupling frequency and where deviations from Hartmann-Hahn condition occur are problems …*

   GH: This sentence was replaced.

39. SI, page 7, eq. (S15) and (S16): What is $q$?

   GH: The in-line equation $q = \sqrt{a^2 + b^2}$ has been inserted in the line after Eq. (S14).

40. SI, page 8, line 1 after 3.3.2.1 Here and elsewhere: replace all *Equ.* by *Eq.*

   GH: Corrected.

41. SI, page 8, line 2 after (S17): replace *what* by *which*

   GH: Corrected.

42. SI, page 9, (S21): typo, change to *crossing*

   GH: Corrected.

43. SI, page 10, line 7 after (S23): Do you mean *approaches* instead of *approximates*?

   GH: Yes; this has to be corrected.

44. SI, page 10, line 7 after (S23): Avoid starting the sentence with I.e.,one could write *In other words, it describes …*

   GH: Corrected to "In other words"

45. SI, page 10, line 3 before 3.3.3.2: Missing word: *The constant component is subject …*

   GH: Corrected.

46. SI, page 12, line 1: *approach* instead of *approximate*

    GH: Corrected.

---

## Author Response (AR1)

**Revision report**

This revision report is written in the form that the reviewer's comments are marked "RC1"; my replies are marked "GH". The changes I have made are indicated by "Changes". All line numbers mentioned in this report refer to the originál manuscript.

**Reply to comments of reviewer 1 and changes in the manuscript**

**RC1**: My major criticism of the paper is that, in its current form, I cannot think of the intended target audience for this paper. If the work is intended as an extension to the typical product operator formalism, which is popular among those who are new to the field, I believe it lacks sufficient background information and explanation of the various cases that are examined to be helpful to that community - I believe it would be more worthwhile to examine fewer cases in the main text for each dimensionality but discuss them more thoroughly. At this level, analytical expressions can be useful to talk about pulse sequence elements, but they quickly lose their applicability as soon as the experimental complexity becomes more than a few pulses. Furthermore, the use of analytical expressions is difficult to motivate in the time when running these simulations (which in this paper correspond to, at largest, 4-dimensional matrices in the Hilbert space) is trivial. Hence, I believe that it could be expanded into an excellent continuation of what is introduced in the conventional product operator formalism.

**GH**: The latter is indeed the intention of the original manuscript. I will emphasize this fact more strongly in the revised manuscript. I also agree that some of the examples should be discussed in more detail. The target audience is researchers who need some insight into the different processes within the experiments. Of course, if someone is only interested in the resulting state of the spin system after a pulse sequence, a numerical calculation will suffice. Sometimes, however, you want to take a look inside this black box. The desire or need to do so could be a motivation to deal with analytical contexts.

**Changes**:

(1) I have inserted the following paragraph in line 17 after '… physical intuition than'.:

a numerical procedure which can seem like a black box. The desire or need to take a look inside this black box could be a motivation to deal with analytical contexts.

(2) The paragraph which ends at line 37 has been extended by the following sentences:

The latter can be seen as an extension of the product-operator formalism to somewhat more complex situations. (i) and (ii) can be useful for simplifying some calculations by dimension reduction without any approximation. Even for numerical calculations this can be useful if it helps to work on low-dimensional systems. As an example, an IS spin system is mentioned here, which is subject to the dipolar interaction, but which is decoupled by rf irradiation, and at the same time fast sample rotation (MAS) modulating the dipolar oscillation takes place. Its time evolution can be described by a system of 3 differential equations using the method presented here. The application of the Shirley-Floquet method will be greatly simplified here. After all, the application of the Liouville-vonNeumann equation in the four-dimensional wavefunction space leads to a system of 16 differential equations even if some of its coefficients can be zero.

(3) The paragraph in line 225/6 has been replaced by

*FID of an ensemble of isolated pairs of equal spins* $\hat{I}_{1x}; \hat{I}_{2x}$ *after a π/2 pulse; homonuclear dipolar interaction within the spin pairs; we observe the transversal magnetization represented by the operator sum* $\hat{I}_{1x} + \hat{I}_{2x}$ *:*

(4) The paragraph in line 228/9 has been replaced by

*FID of an ensemble of isolated pairs of unlike spins (I, S) after a π/2 pulse in the I channel; heteronuclear dipolar interaction within the spin pairs; we observe the transversal I magnetization represented by the operator $\hat{I}_x$ :*

(5) The paragraph in line 231 has been replaced by

*FID of an ensemble of spins I=1 (e.g., $^2H$ or $^{14}N$) under quadrupolar interaction; we follow again the transversal magnetization:*

(6) The paragraph in line 233 has been replaced by

*Ensemble of pairs of homonuclearly coupled equal spins ($I_1$, $I_2$) with spin quantum number 1/2 where initially spin 1 is oriented parallel to $B_0$ and spin 2 antiparallel to that. We follow the difference z magnetization which will be represented by $\hat{I}_{1z} - \hat{I}_{2z}$ :*

(7) The paragraph in line 233 has been replaced by

*Cross polarization within pairs of antiparallel unlike spins (I, S): Both spins are locked in resonant rf fields with equal nutation frequencies $\omega_{1I} = \omega_{1S} \gg D_{IS}$ (Hartmann-Hahn condition). The Hamiltonian and the state operators are given in the doubly-rotating frame following Hartmann and Hahn (1962) where the z direction is along the rf iradiation. If initially the S spins are oriented parallel to the locking field and the I spins are antiparallel to that, the time evolution can be described by following $\hat{I}_z - \hat{S}_z$ :*

**RC1**: I also think that the method is interesting to those who do numerical simulations of spin systems, as this lays out a dimensionality reduction technique that requires no approximation of the system. I acknowledge that this is no longer a topic in the direction of analytical expressions, but I believe it is a paper that is well laid-out for a theoretician in magnetic resonance that would guide them through a dimensionality reduction approach. A section that discusses this possibility might be of interest, although I don't think it is required for publication.

**GH**: For the revised version, I will add this topic to the conclusion section.

**Changes**: The second of the changes due to the previous reviewer's comment also relates to this comment. I have also added the following sentences to the conclusion section:

*In the N-dimensional wavefunction space, the Liouville-vonNeumann equation corresponds to a system of $N^2$ differential equations, which is usually significantly larger than that obtained by the dimension reduction with the method presented here. It may be advantageous to first apply this method on an analytical basis before starting the numerical implementation.*

**RC1**: Broadly, I would like to also ask the author to comment in the article about when this technique can be applied in general, as practical applications are limited to A) Hamiltonians that are time-independent or B) Hamiltonians that can be cast in a frame where they become time-independent, either by use of Average Hamiltonian Theory or other analytical methods like toggling/rotating/interaction frames.

**GH**: This technique is not restricted to time-independent Hamiltonians! The commutator equations are the same if the Hamiltonian is time-dependent. The problem is "simply" to solve the Liouville-vonNeumann equation, which is now a system of linear differential equations, but with non-constant coefficients. No general solution exists. The matrix exponential only corresponds to a first-order Magnus expansion, a

more or less adequate approximation. Nevertheless, some of such equations can be immediately integrated. I also obtained analytical solutions for some cases (for example for dipolar decoupling under MAS), but I did not include them in this manuscript because they seemed to me to be beyond the scope of this article. However, it may be mentioned as an outlook or for further applications.

It should be noted that numerical methods are also sometimes associated with problems. For example, convergence problems may arise in Floquet calculations when the rf strength approaches the spin rate. On the other hand, an analytical solution of the problem could make use of the mathematical properties of the associated function when the parameters approach critical regions, i.e. make use of previous results of mathematics instead of investigating the power in own numerical research.

So in a revised version, I will add a paragraph on time-dependent Hamiltonians.

**Change**: (1) I have added another section dealing briefly with time-dependent interactions which lead to time-dependent Hamiltonians and Liouvillians because I also noticed the possible misunderstanding that this procedure is only applicable to time-independent interactions alone. Of course the third step – matrix exponentialisation – is not applicable here in many cases; the system of differential equations which is still linear but with time-dependent coefficients, has to be solved in a different way to get a propagator matrix or a propagation formula.

(2) I have reworded the last sentence of the introduction and added another sentence referring to the new section (5) on time-dependent interactions.

**RC1**: During the process of writing this report, I do have a following major concern with the formulation as it is written. In equation 19, the author presents the formulation of the product $\mathbf{L} \cdot \boldsymbol{\rho}$ as the left-handed multiplication of the Liouvillian with the basis vector of operators. This led to my comment that equation 22 has index typos in it, as the dot product $\mathbf{U} \cdot \boldsymbol{\rho}$ (also left handed by integration of eq. 19) would generate terms such as $\hat{A}_2 \, U_{12}$ in the evolution of $\hat{A}_1$. In equation 35, the author says that the propagation rules are obtained from the columns of the propagator, however that is not the case if the author has formulated the theory with left-handed multiplication. The propagation rules are obtained from the rows of the propagator. At the end of the day, this is only rectified by the fact that the timesymmetry of quantum mechanics allows for this (up to a definition of a phase). As such, either I have misinterpreted how the evolution is calculated ($\mathbf{U} \cdot \boldsymbol{\rho}$), which is unclear given the ambiguous notation in the appropriate sections, or the author has accidentally taken the wrong set of elements from the propagator.

**GH**: At this point, I do not agree with the reviewer comment. I have to admit that this problem also gave me a headache for some time. Now, for my opinion, the notation in Eq. (22) is correct. Please look at the example at the end of this reply.

The reason for this at first sight counterintuitive behaviour is that we are dealing with two spaces, one of which is the dual space of the other, both describing the same phenomenon. One space contains the "normal" density matrix operations and the propagator matrix action, the other contains the commutator equations and the propagation formulae. A matrix from one space is the transpose of the corresponding matrix in the dual space. This explains why the columns and not the rows of the propagator matrix give the propagation formulae.

**Change**: I added the new subsection 2.3, which explains the two different notations: (i) superoperatormatrix-density *matrix* notation, which belongs to the matrix space $C^d$, and (ii) propagation formulae (those with an arrow) for the density *operator*, which belongs to the operator space. This is the problem of dual spaces; the matrices of that are transposes of each other.

Already in the original version of the SI there is a proof explicitly showing that the coefficient matrix of the propagation formula is the transpose of the propagator matrix in the matrix space. However, I have extended this proof to include the transpose relation between the Liouvillian and coefficient matrix of the commutator equations.

**RC1**: Specifically, I note the following points:

Page 2 line 31: "dipol-dipol" is a typo.

**GH**: This will be corrected.

**Change**: Replaced by dipole-dipole, in line 32 as well.

**RC1**: Page 4 line 79: Minor, but the author should use proper typesetting for dot products ($\mathbf{A}^{\dagger} \cdot \mathbf{B}$ instead of $\mathbf{A}^{\dagger} . \mathbf{B}$). This is a problem throughout page 4, but should be consistent in the entire article.

**GH**: This will be corrected.

**Change**: All full stops which were originally used as symbols for matrix products were replaced by central stops.

**GH1**: Page 5 line 122: "on each operator A" missing the hat on the $\hat{A}$.

**GH**: This will be corrected.

**Change**: Corrected.

**RC1**: Page 6 line 152: I find the notation that is introduced overly confusing for this section, particularly, terms such as the $\lambda_{11}\hat{A}_1$ term in equation 16 are by definition zero, a point that is made in the very line in question. Thus it leaves the reader somewhat confused as to why equation 16 would contain this term. The same can be said of the $\lambda_{22}\hat{A}_2$ term in equation 17. The author should either specifically state that the coefficients $\lambda_{nn}$ by definition or drop them from the equations.

**GH**: $\lambda_{11}$ does not vanish in every case, only for a Hermitian basis. Even though this is true for most but not all examples, I have chosen a formulation that really covers all cases. In the literature (Slichter, Abragam), the density matrix is characterized also by non-hermitian operators, mostly by $\hat{I}_{\pm}$. At the moment I do not see a simpler way to present the procedure in the most general form, but I will think about it.

**Change**: No change, because the $\lambda_{nn}$ do not generally vanish.

**RC1**: Page 7 line 77: "The action of the Liouvillian on any $\hat{A}_2$ ($i \in 1..N$) leads to a linear combination of the $\hat{A}_i$". Firstly, I believe there is a parentheses that is missing in after the N.

**GH**: To be corrected (together with an insertion of curly brackets for denoting a set).

**Change**: Corrected, additionally curly brackets inserted.

**RC1** (continued): However, this is a general result of the operators for spin systems being part of a Lie algebra and is well known. For a detailed explanation, I recommend Spin by Ilya Kuprov.

**GH**: What was meant was "leads to a linear combination of this limited set of A_i (not the whole basis)". This is used to justify the following sentence. I will make this paragraph more explicit.

**Change**: I have changed the end of this sentence to "… to a linear combination of this limited set of the $A_i$."

**RC1**: Page 8 line 196: The author has typos in the indices in equation 22. For instance, one should find the term $\hat{A}_2 \cdot U_{12}$ in the evolution of $\hat{A}_1$ (line 196), if I have interpreted correctly that the dot product being calculated is $\mathbf{U} \cdot \left( \hat{A}_1 \quad \hat{A}_2 \quad … \quad \hat{A}_N \right)^\top$, which is also unclear as it is not stated and only indicated through the ambiguous arrow with a $\hat{H} \cdot t$ decorated over it. Furthermore, please only use dot products when they are actually between two objects of rank-1 or higher, otherwise it is ambiguous what is intended.

**GH**: All dots which do not belong to matrix products will be removed.

**Change**: Corrected in this way. Concerning the index problem: See my reply above; my original indexing is for my opinion correct.

**RC1**: Page 8 line 198: "recompose the matrix exponential for each new situation". It is unclear what the author means by "each new situation". Please elaborate or be specific.

**GH**: I will replace this expression by "… for each new experimental situation. The mathematical structure of the commutator equations is the same for all 3D cases, and the same for all 4D cases. Therefore, the propagators also have an equivalent structure. This means that we can use the generic propagation formulae as a template."

**Change**: The replacement was made as indicated in my reply.

**RC1**: Page 9 line 227: The author introduces in the case of a 2D subspace the operator basis

$$\left\{ \hat{A}_i \right\} = \left\{ \left( \hat{I}_{1x} + \hat{I}_{2x} \right), 2\left( \hat{I}_{1z}\hat{I}_{2y} + \hat{I}_{1y}\hat{I}_{2z} \right) \right\}$$

which is a linear combination of the operators that one would typically encounter when exploring a 2spin system for the first time (formed by the permutations of { $\hat{E}$, $\hat{I}_x$, $\hat{I}_y$, $\hat{I}_z$}). As such, those unfamiliar with why those operators may be linearly combined would likely be confused by this, as it is a nonintuitive basis to build. The author should discuss how, when this procedure is carried out in the native operator basis where the elements are instead

$$\left\{ \hat{A}_i \right\} = \left\{ \hat{I}_{1x}, \hat{I}_{2x}, 2\hat{I}_{1z}\hat{I}_{2y}, 2\hat{I}_{1y}\hat{I}_{2z} \right\}$$

how the appropriate matrices appear and how one can further reduce this 4D case to a 2D case. This is a critical part of the procedure that is missing from the article, and is not unique to this instance.

**GH**: I will add a note to the revised version that this low-dimensional subspace will result from the process. The non-intuitivity is something like the price for the low dimensionality. I will also check the other examples for the need for similar additional explanations.

**Change**: After the end of line 238, I inserted the following sentences:

*These examples show an effect of the dimension reduction: To obtain a 2D problem, the operators characterizing the states of the spin system have a more complicated structure than in the simple cases above. For example, it would be possible, to consider the right side of Eq. (32) as a linear combination of the four states $\hat{I}_{1x}, \hat{I}_{2x}, 2\hat{I}_{1z}\hat{I}_{2y}$, and $2\hat{I}_{1y}\hat{I}_{2z}$, if a more illustrative notation is desired.*

**RC1**: Page 10 line 235: If the author wishes to discuss the case of cross polarization, it should be noted that the author has rearranged the initial spin state into $\hat{S}_z - \hat{I}_z$ and $\hat{S}_z + \hat{I}_z$, the latter of which does not evolve and the former which dictates the polarization transfer. This goes along the lines of explaining the operator basis that is used, as without this, the problem would be at least 3D (if one already has collected the zero-quantum terms into one basis state).

**GH**: Yes, it should be shown that the polarization difference is an appropriate variable to have a situation that is POF-like, hence 2D. I will add a note that using single polarizations would lead to a 3D case.

**Change**: After Eq. (36) (revised version), the following sentence is added:

*The last equation describes the behavior of the difference of I and S polarization, not the individual polarizations themselves. The time evolution of the latter requires at least a 3D approach, see below.*

**RC1**: Page 10 lines 241: "The procedure described above reaches the cancellation condition after three commutators of the kind". This language is ambiguous and makes it sound like equation 33 is a generic result to the method and not specific for the types of systems that belong to the 3D case.

**GH**: The sentence should begin with "Here we are dealing with those cases where the procedure described above reaches… "

**Change**: The original sentence has been replaced by: "Here we are dealing with those cases where the procedure described above reaches the cancellation condition after three commutator equations of the kind…".

**RC1**: Page 10 line 246: It would be helpful if the author showed how this was calculated.

**GH**: I will add a note that the propagator matrix has been calculated from the Liouvillian matrix as matrix exponential corresponding to Eq. (20). This can be done for example by using MATHEMATICA, see also SI. More technical details concerning matrix exponentialization are out of the scope of this manuscript.

**Change**: Line 245 has been changed to "and from that the matrix of the superpropagator as matrix exponential corresponding to Eq. (20):"

**RC1**: Page 12 line 292: "In resonance" should be "on resonance".

**GH**: This will be corrected.

**Change**: Corrected.

**RC1**: Page 13 line 296: "In resonance" should be "on resonance".

**GH**: This will be corrected.

**Change**: Corrected.

**RC1**: Page 14 line 321: It would be helpful to explain what the LG condition is to the reader.

**GH**: This sentence will be replaced by:

The interaction between $S_1$ and $S_2$ is assumed to be zero. This can be realized experimentally by irradiating the S spins with a resonance offset which is $1/\sqrt{2}$ times the rf strength which is denoted by "Lee-Goldburg condition", see [citation].

**Change**: This sentence has been replaced by:

*The interaction between $S_1$ and $S_2$ is assumed to be zero. This can be realized experimentally by irradiating the S spins with a resonance offset which is $1/\sqrt{2}$ times the rf strength, known as Lee-Goldburg condition, see Lee and Goldborg (1965).*

**RC1**: Page 15 line 341: "the more the smaller the rf power" clumsy phrasing.

**GH**: This sentence will be reworded in the revised manuscript.

**Change**: This sentence has been replaced by

*Contrary to the relations obtained for infinite $\omega_1$, the sum of both polarizations oscillates. The amplitude decreases with increasing rf power.*

**Reply to comments of reviewer 2 and changes in the manuscript**

**RC2**: To make the manuscript easier to understand for newcomers I suggest to include a bit more background information. Some essential background knowledge is taken for granted, making the understandability sometimes a bit difficult. In my opinion, it should be explained explicitly what is behind the "arrow notation" of the propagation rules introduced in equation (1) and (2), for example, by providing an equation like

$$\hat{\rho}(t) = \exp\left(-i\hat{H}t\right)\hat{\rho}(0)\exp\left(i\hat{H}t\right)$$

By expanding the exponentials in this equation the occurence of the multiple commutators can be readily explained. Without this, the mentioning of repeated calculations of commutators comes "out of the blue", at least for less experienced readers.

**GH**: I will mention in the introduction that the arrow notation is a widely used notation at least if the product-operator formalism is applied.
Moreover, as a consequence to the discussions concerning the indices in Eq. (22) and the related SI section 1.1, I will insert into 2.1 a paragraph dealing with the connection between the propagation rules (those

with an arrow) and the superoperator-density operator equations like $\hat{\rho}(t) = \hat{\tilde{U}} \hat{\rho}_0$. This also serves to emphasise that the two forms of the representation of the time evolution take place in different spaces: The arrow notation uses an operator base, while the matrix notation uses a base of column matrices. This means that the corresponding matrices are transposed to each other because they originate from dual spaces. I hope that this will clear up the misunderstandings regarding the indexing in equation (22).

My access to the commutator equation systems comes from the requirement of having a Liouville-invariant subspace, i.e. a multiple application of the Liouvillian and therefore the commutator serves as a tool to ensure the L-invariance of the subspace. Then this criterion is used also to find such a subspace. I will emphasize this more in the subsections 3.1 and/or 3.2.

Changes:

(1) As already mentioned in more detail as a change with reference to the corresponding comment of reviewer 1, I have added the new subsection 2.3, which explains the two different notations used here (density matrix notation as equation vs. propagation formula (arrow formulation)).

(2) I have added the following sentence in line 173 after the full stop:

*Consequently, the search for this Liouvillian-invariant minimum subspace is performed by repeatedly calculating the commutator of the Hamiltonian with the operator representing the initial state of the spin system, as shown in detail below.*

**RC2**: I also have a problem understanding section 1.1 of the SI: I find it difficult to bring the equation and the text above it together — perhaps the text can be rephrased more clearly. I guess the meaning of the arrow $7\rightarrow$ is *maps to*.

**GH**: I agree that this section of the SI was inadequate to explain some relations from the Main Part. The purpose of this section is to prove that (i) the Liouvillian matrix is the transposed coefficient matrix of the system of commutator equations, and (ii) the coefficient matrix of the propagation formulae is the transposed propagator matrix. I have rewritten this section completely in the form containing proposition and proof. I hope that this will enable the reader to see the purpose of the section and, most importantly, to see why some relations given in the Main Part exist. For the latter, I will insert some remarks at the relevant places in the Main Part.

**Changes**:

(1) Section SI 1.1 has been completely redrafted according to my reply.

(2) References to SI 1.1 have been inserted where relevant in the main text. This concerns the following lines: Immediately below Eq. (23), between Eqs. (25) and (26), above Eq. (40), and above Eq. (58) (these equation numbers refer to the revised manuscript).

**RC2**: It may also help to explicitly explain the different symbols used for abstract operators (such as $\hat{\rho}$) and their matrix representations (such as $\rho$).

**GH**: I will insert a paragraph explaining these different symbols.

**Change**: The following sentences have been added at the end of the introduction (line 51):

*As usual, in this manuscript operators are denoted by a hat ($\hat A$) and superoperators by a double hat ($\hat{\hat B}$), while the vector or the matrix associated with this operator is denoted by the same symbol but in bold style and without hat ($\bf A$, $\bf B$). Scalar variables are written in italics.*

**Technical issues**

1. **RC2:** page 1, line 22ff: The introductory example mentioned here is propagation of transversal magnetization of spin $I = 1/2$ (which is a good choice) but the following equation (1) and the text on page 2, line 25, contain $I_z$ instead of $I_x$. In eq. (1), both occurrences of $I_z$ should be replaced.

   **GH**: Thank you, this has been corrected.

   **Change**: This has been corrected

2. **RC2:** page 2, line 28: I suggest to replace *in this case* by *in this example*.

   **GH**: This suggestion has been added.

   **Change**: This suggestion was followed.

3. **RC2:** page 2, line 31: Typo in *dipol-dipol*

   **GH**: Corrected.

   **Change**: Corrected.

4. **RC2:** same line: I suggest to add *or* before *cross polarization*.

   **GH**: This suggestion has been added.

   **Change**: 'or' has been added.

5. **RC2:** page 3, lines 50–51: I suggest to replace *independent of the dimension of the latter* by *although the Liouville space has a much larger dimension*.

   **GH**: This suggestion has been added.

6. **RC2:** page 3, line 52: I wonder if the statement *"However, condition (3) cannot be fulfilled if more than one interaction has to be considered"* is always true.

   **GH**: I know of no counterexample, but I know of no proof. So I replace the "cannot be" with "is often not".

   **Change**: 'cannot be' is replaced by 'is often not'.

7. **RC2:** page 3, line 54: *...an initial state $\rho_0 = \hat{I}_z$.* (Shouldn't the density operator carry a ˆ?). This is one of the few instances, where the equal sign (=) is used for assigning the initial state. In most of the manuscript (including SI), assignments of special values are indicated by arrows ($\rightarrow$). I prefer the equal sign because the arrow can be misinterpreted as indication of a limiting value or, in the context of this manuscript, a time evolution.

   **GH**: I agree with that comment. At first I wanted to characterize a substitution with this arrow, but now I see the danger of confusion and will replace all arrows with equal signs. The missing hat has been added.

   **Changes**: (1) The hat was added to $\rho_0$, (2) all assignment arrows are replaced by equal signs.

8. **RC2:** page 3, line 63: I suggest to rephrase the sentence (for better understandability) and write: *…note the 2 × 4 matrix in Eq. (5) is the exponential of the 2 × 2 matrix in Eq. (6) multiplied by −it ….*

   **GH**: I have changed my inadequate formulation to this suggestion.

   **Change**: I replaced this sentence by

   *For this analysis, it is important to note that the 2x2 matrix in Eq. (5) is the exponential of the 2x2 matrix in Eq. (6) multiplied by -it (see SI, section 2):*

9. **RC2:** page 3, line 67: change *formed* to *formulated*

   **GH**: Changed.

   **Change**: Changed to 'formulated'.

10. **RC2:** page 3, line 70: change *was possible* to *is possible*

    **GH**: Changed.

    **Change**: Changed to 'is'.

11. **RC2:** page 3, line 74: change *for the further work here* to *this work*

    **GH**: Changed.

    **Change**: Corrected as suggested.

12. **RC2:** page 4, line 85: I think *estimating* should be changed to *calculating*. (There are more instances, where *estimate* is used instead of *calculate*. Please check.)

    **GH**: Corrected.

    **Change**: Corrected as suggested.

13. **RC2:** page 4, lines 86–87. I suggest to rephrase the sentence: *…but it depends on the relevant space, which is different for different numbers of spins.*

    **GH**: This suggestion has been added.

    **Change**: This suggestion has been added.

14. **RC2:** page 5, line 118: change estimation to *calculation*

    **GH**: Corrected.

    **Change**: Corrected as suggested.

15. **RC2:** page 5, line 122: The operator $\hat{A}$ is missing its hat.

    **GH**: Corrected.

    **Change**: The hat has been added.

16. **RC2:** page 6, lines 152–153: *see Example 1D-1 in the SI.* In the SI, there is no such example. A 1D subspace is mentioned in section 4.1.

    **GH**: Yes, that needs to be corrected.

    **Change**: *Example 1D-1 i n the SI* has been replaced by *subsection 4.1.*

17. **RC2:** page 7, line 172: extra *all*

**GH**: Corrected.

**Change**: The extra all has been removed.

18. **RC2:** page 7, line 177: parenthesis not closed

    **GH**: Corrected.

    **Change**: Parentheses closed.

19. **RC2:** page 7, line 181: extra *above*

    **GH**: Corrected.

    **Change**: Extra *above* removed.

20. **RC2:** page 7, lines 181–182: Shouldn't all $N$ be replaced by $n$?

    **GH**: Yes, that needs to be corrected.

    **Change**: I have replaced all $n$ by $N$.

21. **RC2:** page 8, Eq. 22: Isn't the matrix U multiplied from the left, resulting in $\hat{A}_1 \cdot U_{11} + \hat{A}_2 \cdot U_{21} + ...$ (inverted indices of $U_{kl}$)?

    **GH**: The coefficient matrix of the propagation formulae is the transposed propagator matrix, so the indices in Eq. (22) are correct. This relationship was not adequately mentioned in the original manuscript incl. SI. See the reply to the reviewer's comment on SI section 1.1 above. This SI section has been completely rewritten to emphasize this relationship.

    **Change**: I added in line 196 after ‚elements of the‘: *transposed.*

22. **RC2:** page 9, line 219: I suggest *...appearing in Eq. (5) and (6).*

    **GH**: This suggestion has been added.

    **Change**: This insertion has been made accordingly.

23. **RC2:** page 10, line 243: replace *estimate* by *calculate*

    **GH**: This sentence has been changed due to insertion of a note that the Liouvillian is the transposed coefficient matrix of the commutator equations.

    **Change**: New sentence: *In step 2, we determine the Liouvillian matrix as transposed coefficient matrix of Eq. (37):*

24. **RC2:** page 12, line 286: typo, it should probably read: *...with the amplitude $D_{IS}^2 \big/ \left( \omega_{IS}^2 + D_{IS}^2 \right)$*

    **GH**: Corrected.

    **Change**: Expression corrected as suggested.

25. **RC2:** page 13, line 319: replace DRKS by *doubly rotating frame* (I think it should be "doubly rotating" instead of "double rotating" everywhere.)

    **GH**: Corrected everywhere.

    **Change**: This replacement has been done.

26. **RC2:** page 17, line 367: I think *...larger prefactor, which reflects the roof effect* is correct.

**GH**: Corrected.

**Change**: Corrected in the suggested way.

27. **RC2:** page 18, line 376: *a limited power*

    **GH**: Corrected.

    **Change**: Corrected as suggested.

28. **RC2:** page 20, line 406: *an I spin*

    **GH**: Corrected.

    **Change**: *a* replaced by *an*.

29. **RC2:** page 21, lines 432–433: *N* and *n* not clear. I think *N* is the total number of spins, and *n* the number of factors in the product. For clarity one should write $2^{(N/2)-n}$—if I understood it correctly.

    **GH**: *N* and *n* have the meaning that the reviewer assumed. I change the sentence to include an explanation for these variables. The N/2 in the exponent is set into parentheses.

    **Change**: I have been inserted before this sentence: *Consider a system of N spins 1/2.*

    Moreover, I have replaced the *1/2* before the paranthesis by: *from this N-spin system*

30. **RC2:** page 22, line 461: What is $\omega_{1I;S}$?

    **GH**: I have replaced this variable with mixed index by two separate terms $\omega_{1I}$ and $\omega_{1S}$.

    **Change**: $D_{IS} \ll \omega_{1I;S}$ has been replaced by $D_{IS} \ll \omega_{1I}, \omega_{1S}$

31. **RC2:** page 23, lines 469–470: Perhaps better *...developed the method, used it to derive the examples given here and ...*?

    **GH**: This proposal was adopted.

    **Change**: Done as proposed.

32. **RC2:** page 23, line 472: no plural for *advice*

    **GH**: Corrected.

    **Change**: Corrected.

33. **RC2:** SI, page 4, line after (S2): instead of *Similarly* the use of *Similar to the dipolar Hamiltonian* might be more informative.

    **GH**: This proposal was adopted.

    **Change**: Replaced as proposed.

34. **RC2:** SI, page 4, line 4 from bottom: How about *...is parallel magnetization of spins $I_1$ and $I_2$, aligned transversal to* $\mathbf{B_0}$?

    **GH**: This proposal was adopted.

    **Change**: Replaced as proposed.

35. **RC2:** SI, page 5, line after 3.2.2.3: The extra punctuation mark after *Hamiltonian:* should be deleted.

    **GH**: Corrected.

**Change**: Extra punctuation has been removed.

36. **RC2:** SI, page 6, sentence before 3.2.3.2: *...not −(3/2)D$_{II}$...(example 2D-1)* (minus-sign for complete-ness, wrong example number)

    **GH**: The minus is added, the example number is changed to 2D-1.

    **Change**: Done as written in the reply.

37. **RC2:** SI, page 7, line 7 (including eq.): *...can be detected ...*

    **GH**: Corrected.

38. **RC2:** SI, page 7, change of sentence: The cases where the relevant magnetic field strengths are not large with respect to the coupling frequency and where deviations from Hartmann-Hahn condition occur are problems ...

    **GH**: This sentence was replaced.

    **Change**: The whole paragraph has been reworded.

39. **RC2:** SI, page 7, eq. (S15) and (S16): What is *q*?

    **GH**: The in-line equation $q = \sqrt{a^2 + b^2}$ has been inserted in the line after Eq. (S14).

    **Change**: Done as written in the reply.

40. **RC2:** SI, page 8, line 1 after 3.3.2.1 Here and elsewhere: replace all *Equ.* by *Eq.*

    **GH**: Corrected.

    **Change**: All equation references are now written as *Eq.*

41. **RC2:** SI, page 8, line 2 after (S17): replace *what* by *which*

    **GH**: Corrected.

    **Change**: *what* by *which* replaced.

42. **RC2:** SI, page 9, (S21): typo, change to *crossing*

    **GH**: Corrected.

    **Change**: Corrected.

43. **RC2:** SI, page 10, line 7 after (S23): Do you mean *approaches* instead of *approximates*?

    **GH**: Yes; this has to be corrected.

    **Change**: *approximates* replaced by *approaches*.

44. **RC2:** SI, page 10, line 7 after (S23): Avoid starting the sentence with I.e.,one could write *In other words, it describes ...*

    **GH**: Corrected to "In other words"

    **Change**: *I.e.* replaced by *In other words*.

45. **RC2:** SI, page 10, line 3 before 3.3.3.2: Missing word: *The constant component is subject ...*

    **GH**: Corrected.

**Change**: *is* is inserted; the sentence begins now as proposed by reviewer 2.

46. **RC2:** SI, page 12, line 1: *approach* instead of *approximate*

    **GH**: Corrected.

    **Change**: *approximate* replaced by *approach*.

**Other changes, not referring to any comment:**

The abstract and conclusion sections have been modified to reflect the changes, in particular the inclusion of a section on time-dependent problems.

Eq. (19): "≡ L(..)" has been removed.

Line 196: Inserted "transpose" before propagator matrix.

Line 100: The equation in this line has been changed transformed from an inline equation to a numbered equation. As a result, the numbers of the following equations have been increased.
* * *
**Example which demonstrates for a 2D space that the coefficient matrix of the propagation formulae is the transpose of the propagator matrix**

Given a propagator matrix **U** which performs the time evolution in a 2D Liouville space of the initial density column $\boldsymbol{\rho}_0 = \left(\rho_{01}\ \rho_{02}\right)^{\mathsf{T}}$ to that of a later time $\boldsymbol{\rho} = \left(\rho_1\ \rho_2\right)^{\mathsf{T}} = \mathbf{U}\boldsymbol{\rho}_0$ :

$$\begin{pmatrix} \rho_1 \\ \rho_2 \end{pmatrix} = \begin{pmatrix} U_{11} & U_{12} \\ U_{21} & U_{22} \end{pmatrix} \begin{pmatrix} \rho_{01} \\ \rho_{02} \end{pmatrix} \tag{1}$$

Let $\hat{A}$ and $\hat{B}$ be basis operators of this 2D space with the corresponding matrices

$$\mathbf{A} = \begin{pmatrix} 1 \\ 0 \end{pmatrix}; \qquad \mathbf{B} = \begin{pmatrix} 0 \\ 1 \end{pmatrix}.$$

Now let us consider the two cases where (i) $\boldsymbol{\rho}_0 = \mathbf{A}$, (ii) $\boldsymbol{\rho}_0 = \mathbf{B}$ characterize the initial state. The time evolution will be described by

$$\text{(i)} \quad \mathbf{U}\mathbf{A} = \begin{pmatrix} U_{11} & U_{12} \\ U_{21} & U_{22} \end{pmatrix} \begin{pmatrix} 1 \\ 0 \end{pmatrix} = \begin{pmatrix} U_{11} \\ U_{21} \end{pmatrix} = U_{11} \underbrace{\begin{pmatrix} 1 \\ 0 \end{pmatrix}}_{\mathbf{A}} + U_{21} \underbrace{\begin{pmatrix} 0 \\ 1 \end{pmatrix}}_{\mathbf{B}}$$

$$\text{(ii)} \quad \mathbf{U}\mathbf{B} = \begin{pmatrix} U_{11} & U_{12} \\ U_{21} & U_{22} \end{pmatrix} \begin{pmatrix} 0 \\ 1 \end{pmatrix} = \begin{pmatrix} U_{12} \\ U_{22} \end{pmatrix} = U_{12} \underbrace{\begin{pmatrix} 1 \\ 0 \end{pmatrix}}_{\mathbf{A}} + U_{22} \underbrace{\begin{pmatrix} 0 \\ 1 \end{pmatrix}}_{\mathbf{B}}$$

Translated to propagation formulae this becomes

$$
\begin{array}{lll}
\text{(i)} & \mathbf{A} & \rightarrow \quad U_{11}\,\mathbf{A} + U_{21}\,\mathbf{B} \\
\text{(ii)} & \mathbf{B} & \rightarrow \quad U_{12}\,\mathbf{A} + U_{22}\,\mathbf{B}
\end{array}
$$

or

$$
\begin{pmatrix} \mathbf{A} \\ \mathbf{B} \end{pmatrix} \rightarrow \begin{pmatrix} U_{11} & U_{21} \\ U_{12} & U_{22} \end{pmatrix}\begin{pmatrix} \mathbf{A} \\ \mathbf{B} \end{pmatrix}. \tag{2}
$$

Comparing equations (1) and (2), we conclude for this example that the coefficient matrix of the propagation formulae is the transpose of the matrix of the propagator. In other words, the columns of the propagator **U**, not the rows, form the propagation formulae.

∎

**U** and **U**$^{\mathrm{T}}$ describe the same phenomenon in two different spaces where one is the dual space of the other.